# Risk factors during first 1,000 days of life for carotid intima-media thickness in infants, children, and adolescents: A systematic review with meta-analyses

Adina Mihaela Epure[1,2]*, Magali Rios-Leyvraz[2], Daniela Anker[3], Stefano Di Bernardo[4], Bruno R. da Costa[3,5], Arnaud Chiolero[1,2,3,6]☯, Nicole Sekarski[4]☯

1 Population Health Laboratory (#PopHealthLab), University of Fribourg, Fribourg, Switzerland, 2 Department of Epidemiology and Health Services, Center for Primary Care and Public Health (UNISANTÉ), University of Lausanne, Lausanne, Switzerland, 3 Institute of Primary Health Care (BIHAM), University of Bern, Bern, Switzerland, 4 Paediatric Cardiology Unit, Woman-Mother-Child Department, Lausanne University Hospital (CHUV), Lausanne, Switzerland, 5 Institute of Health Policy, Management and Evaluation, University of Toronto, Toronto, Canada, 6 Department of Epidemiology, Biostatistics and Occupational Health, McGill University, Montréal, Canada

☯ These authors contributed equally to this work.
* adina-mihaela.epure@unifr.ch

**Data Availability Statement:** All relevant data and citations to all data sources, that is, the full-text

## Abstract

### Background

The first 1,000 days of life, i.e., from conception to age 2 years, could be a critical period for cardiovascular health. Increased carotid intima-media thickness (CIMT) is a surrogate marker of atherosclerosis. We performed a systematic review with meta-analyses to assess (1) the relationship between exposures or interventions in the first 1,000 days of life and CIMT in infants, children, and adolescents; and (2) the CIMT measurement methods.

### Methods and findings

Systematic searches of Medical Literature Analysis and Retrieval System Online (MED-LINE), Excerpta Medica database (EMBASE), and Cochrane Central Register of Controlled Trials (CENTRAL) were performed from inception to March 2019. Observational and interventional studies evaluating factors at the individual, familial, or environmental levels, for instance, size at birth, gestational age, breastfeeding, mode of conception, gestational diabetes, or smoking, were included. Quality was evaluated based on study methodological validity (adjusted Newcastle–Ottawa Scale if observational; Cochrane collaboration risk of bias tool if interventional) and CIMT measurement reliability. Estimates from bivariate or partial associations that were least adjusted for sex were used for pooling data across studies, when appropriate, using random-effects meta-analyses. The research protocol was published and registered on the International Prospective Register of Systematic Reviews (PROSPERO; CRD42017075169). Of 6,221 reports screened, 50 full-text articles from 36 studies (34 observational, 2 interventional) totaling 7,977 participants (0 to 18 years at CIMT assessment) were retained. Children born small for gestational age had increased CIMT (16

publications of included studies, are within the manuscript and its supporting information files.

**Funding:** This work was funded by the Swiss National Science Foundation (www.snf.ch; project number 32003B-163240; grantee: AC). The funder had no role in study design, data collection and analysis, decision to publish, or preparation of the manuscript.

**Competing interests:** The authors have declared that no competing interests exist.

**Abbreviations:** ART, assisted reproductive technologies; CCA, common carotid artery; CENTRAL, Cochrane Central Register of Controlled Trials; CI, confidence interval; CIMT, carotid-intima media thickness; CVD, cardiovascular disease; DOHaD, Developmental Origins of Health and Disease; EMBASE, Excerpta Medica database; MEDLINE, Medical Literature Analysis and Retrieval System Online; MOOSE, Meta-analysis of Observational Studies in Epidemiology; PRISMA, Preferred Reporting Items for Systematic Reviews and Meta-Analyses; PROSPERO, International Prospective Register of Systematic Reviews; SMD, standardized mean difference.

studies, 2,570 participants, pooled standardized mean difference (SMD): 0.40 (95% confidence interval (CI): 0.15 to 0.64, $p$: 0.001), $I^2$: 83%). When restricted to studies of higher quality of CIMT measurement, this relationship was stronger (3 studies, 461 participants, pooled SMD: 0.64 (95% CI: 0.09 to 1.19, $p$: 0.024), $I^2$: 86%). Only 1 study evaluating small size for gestational age was rated as high quality for all methodological domains. Children conceived through assisted reproductive technologies (ART) (3 studies, 323 participants, pooled SMD: 0.78 (95% CI: −0.20 to 1.75, $p$: 0.120), $I^2$: 94%) or exposed to maternal smoking during pregnancy (3 studies, 909 participants, pooled SMD: 0.12 (95% CI: −0.06 to 0.30, $p$: 0.205), $I^2$: 0%) had increased CIMT, but the imprecision around the estimates was high. None of the studies evaluating these 2 factors was rated as high quality for all methodological domains. Two studies evaluating the effect of nutritional interventions starting at birth did not show an effect on CIMT. Only 12 (33%) studies were at higher quality across all domains of CIMT reliability. The degree of confidence in results is limited by the low number of high-quality studies, the relatively small sample sizes, and the high between-study heterogeneity.

## Conclusions

In our meta-analyses, we found several risk factors in the first 1,000 days of life that may be associated with increased CIMT during childhood. Small size for gestational age had the most consistent relationship with increased CIMT. The associations with conception through ART or with smoking during pregnancy were not statistically significant, with a high imprecision around the estimates. Due to the large uncertainty in effect sizes and the limited quality of CIMT measurements, further high-quality studies are needed to justify intervention for primordial prevention of cardiovascular disease (CVD).

## Author summary

### Why was this study done?

- Exposure to adverse experiences in the first 1,000 days of life, i.e., from conception to age 2 years, may determine adaptative changes in the blood vessel walls and increased carotid intima-media thickness (CIMT) in infants, children, and adolescents. This implies carotid arteries with thicker walls that may be due to changes in blood flow and pressure or other factors related to the process of atherosclerosis.

### What did the researchers do and find?

- We performed a systematic review of published studies with meta-analyses and included 36 studies, involving 7,977 participants between 0 and 18 years at CIMT assessment.

- Risk factors in the first 1,000 days of life, particularly poor fetal growth, are associated with increased CIMT in infants, children, and adolescents.

- There is scarce evidence from interventional studies beginning in the first 1,000 days, and none was shown to prevent or improve vascular remodeling in children.

- CIMT measurement protocols in children are heterogeneous and often poorly reported.

### What do these findings mean?

- From a public health perspective, acting early in life by preventing risk factors such as poor fetal growth could help maintain a low cardiovascular risk over the life course.

- Assessing vascular structure and function in children is important to better characterize lifetime risk trajectories and tailor primordial prevention of cardiovascular disease (CVD). Primordial prevention aims to prevent the development of risk factors instead of treating them.

- From a clinical standpoint, promotion of a healthy lifestyle is important at any age, and screening of postnatal cardiovascular risk factors, targeted at children exposed to risk factors in the first 1,000 days of life, may be warranted.

- A widely accepted standardized CIMT measurement protocol in children is needed.

## Introduction

### Background

Within a Developmental Origins of Health and Disease (DOHaD) framework, the first 1,000 days of life, i.e., the period from conception to age 2 years, is regarded as a critical period for long-term cardiovascular health [1]. Evidence suggests that adaptations in body structure and function during development, as a response to cues from the environment, for instance, assisted reproductive technologies (ART) for conception [2], undernutrition, or environmental pollutants [1], could increase the risk for cardiovascular disease (CVD). Indeed, several studies showed that a low birth weight is associated with higher blood pressure [3] or diabetes risk [4], as well as with cardiovascular mortality in adulthood [5].

Carotid intima-media thickness (CIMT) is a marker of CVD risk linked to pathways that originate in the first 1,000 days of life. In adults, CIMT is predictive of heart attack and stroke risk [6,7] and is also associated with prenatal risk factors, such as impaired fetal growth and preterm birth [8,9]. Within a DOHaD perspective, these associations may be explained by early life adaptative changes in the vascular phenotype [10], with lifelong effects on the CVD risk. Nonetheless, in children, the evidence on increased CIMT after impaired fetal growth or preterm birth is inconsistent [11–14]. Further, CVD risk is determined by a combination of factors, at multiple levels, rather than a single factor [15]. A large range of early life factors at the child, familial, or environmental levels needs to be explored to better understand CIMT changes in children. To our knowledge, such a comprehensive overview is lacking.

CIMT assessment in children can be challenging [16,17]. Several aspects related to the ultrasound equipment [18–20], the site of measurement, or the edge detection approach may influence the reliability of measurements [17,21]. Further, measurements in infants and young children are challenging due to limited compliance, anatomic particularities, and a lack of recommendations tailored to this age group [16,22]. The comparability of studies is therefore limited, and the inconsistency in findings may be attributed, at least in part, to the heterogeneity in CIMT measurement. An appraisal of the CIMT measurement methods in children is needed to shed light on these limitations.

We therefore aimed to perform a systematic review with meta-analyses to (1) assess the relationship between exposures or interventions in the first 1,000 days of life and CIMT in infants, children, and adolescents; and (2) critically appraise the CIMT measurement methods.

## Methods

### Protocol development and reporting

The protocol for this systematic review was registered on the International Prospective Register of Systematic Reviews (PROSPERO; CRD42017075169) and published in full in a peer-reviewed journal [23]. The reporting of this paper complies with the Preferred Reporting Items for Systematic Reviews and Meta-Analyses (PRISMA) guidelines [24] (S1 PRISMA Checklist) and the Meta-analysis of Observational Studies in Epidemiology (MOOSE) guidelines [25].

### Eligibility criteria

**Study designs.** Observational and interventional studies with the following designs were considered for inclusion: cohort, case–control, cross-sectional studies, randomized trials, non-randomized controlled trials, and noncontrolled trials. Case reports, case series, opinion papers, letters to the editor, comments, conference abstracts, policy papers, reviews and meta-analyses, study protocols without baseline data, and animal studies were excluded.

**Participants.** We considered for inclusion studies in children from birth up to 18 years old, including studies where participants were recruited pre-birth. Apparently, healthy children and participants with clinical conditions relating to the first 1,000 days of life (e.g., prematurity) were included. Studies in children with rare or special conditions, such as congenital heart diseases, or postnatal conditions conferring an increased risk of CVD, such as hypertension or diabetes, were not considered for inclusion. We aimed to make inferences to the general population. Further, we assumed that there may be a different effect of DOHAD exposures in these groups than in the general population, which should be evaluated separately.

**Exposures/Interventions.** We considered for inclusion prenatal and postnatal exposures, at the individual, familial, or environmental level, which reflected the developmental milieu of the child and occurred between conception and age 24 months (i.e., first 1,000 days of life). These included biological factors related to physiology, physiopathology, or epigenetics and contextual factors related to socioeconomic status, behaviors, or environmental toxins. The selection of exposures was informed by the review of Hanson and Gluckman [1] and Moore and colleagues [26] and was conditioned by the timing of occurrence or ascertainment of exposure, as appropriate. Exposures that started in the first 1,000 days of life, but continued in later life, were included if they spanned up to 5 years of age. Likewise, periconceptional factors that reflected the environment in which the baby was conceived and grew were included. For interventional studies, we considered both pharmacological and non-pharmacological interventions, at the individual, familial, or environmental level, conducted in the first 1,000 days of life. Specifically, eligible interventions had to start and end in the first 1,000 days of life or start in the first 1,000 days of life and continue in later life no longer than child age 5 years.

**Comparators.** Where applicable, the use of a reference group without the exposure or intervention of interest was sufficient for inclusion.

**Outcome measures.** The outcome was the intima-media thickness of the carotid artery measured by ultrasonography from birth to 18 years of age.

**Time frame and setting.** There was no restriction by length of follow-up. There was no restriction by type of setting.

**Language.** Studies in English and French were considered for inclusion.

## Search strategy

Systematic searches were conducted in the Medical Literature Analysis and Retrieval System Online (MEDLINE) database, Excerpta Medica database (EMBASE), and Cochrane Central Register of Controlled Trials (CENTRAL) from inception to 18 March 2019. The strategies for the systematic searches included the 2 main concepts of this systematic review: (1) children and adolescents; and (2) CIMT (detailed search strategies are available in S1 Table). Supplementary searches consisted of (1) a manual search of reference lists and other reviews on the topic; (2) forward citation tracking on Web of Science based on retrieved eligible reports; and (3) personalized search queries in Google Scholar and trial registers. The full strategies are available in S2 Table.

## Study selection process

Following systematic searches in the databases, all retrieved reports were imported into Endnote (version X8.1, Clarivate Analytics, London, United Kingdom), and duplicates were removed according to the method of Bramer and colleagues [27]. Subsequently, reports were uploaded to Covidence (www.covidence.org), a systematic review management platform. Two reviewers performed study eligibility screening independently and in duplicate. Each reviewer screened the reports in a 2-step approach: (1) based on titles and abstracts; and (2) based on full-text of reports retained in the first step. The reviewers agreed on 94% of the titles and abstracts and on 84% of the full texts. Disagreements between reviewers were resolved by discussion, or, if necessary, by a third reviewer. The investigator of 1 completed study (ClinicalTrials.gov identifier: NCT02147457), with unpublished data on CIMT, was contacted by e-mail, but data could not be made available due to technical and logistical issues when performing measurements.

## Data extraction

Data were extracted independently and in duplicate by 2 reviewers using a standardized electronic form in Microsoft Excel (version 2016, Microsoft Corporation, Redmond, United States of America). The data extraction process was piloted on a subset of articles, and a guide for extraction was designed to minimize errors and enhance consistency between reviewers. We extracted information about (1) study characteristics; (2) population characteristics; (3) CIMT definition and additional characteristics of the CIMT measurement methods; (4) characteristics of each exposure or intervention type; and (5) adjusted and unadjusted association or effect estimates.

The quality of each study was evaluated independently by 2 reviewers using an adjusted version of the Newcastle–Ottawa Scale for observational studies [28] and Cochrane collaboration risk of bias tool for interventional studies [29,30] (S7 Table). The quality of the CIMT measurement method was examined according to a predefined tool with 3 levels (higher, lower, and unclear reliability). The complete tool and the algorithm of judgment are available in S3 Table.

Disagreements in extracted data between the 2 reviewers were resolved by discussion or with the arbitration of a third reviewer. Essential missing information was searched by checking additional references related to that study, such as the published research protocol. We also attempted to contact authors of 2 studies by e-mail.

## Data analysis

Analyses were conducted as prespecified in the study protocol [23]. Data transformations and estimations were done according to recommendations and formulae provided in Borenstein

and colleagues [31,32], the Cochrane Handbook for Systematic Reviews [29], Lipsey and colleagues [33], Rücker and colleagues [34], Wan and colleagues [35], Aloe and colleagues [36], and Nieminen and colleagues [37]. The following data transformations were performed: (1) when the mean was not available, it was estimated from the median [35]; (2) when standard deviations were not available, they were estimated from the range or interquartile range [36], standard errors [33], and confidence intervals (CIs) [29,33]; and (3) when standard errors were not provided, they were calculated from CIs, *p*-values, and t-values [29,33]. To perform meta-analyses, additional data simplifications were performed as follows: (1) if a study reported on the same exposure or outcome type at multiple time points, the latest time point was included in the analyses; (2) if a study reported on multiple exposed or reference subgroups, they were combined into 1 using formulae provided by the Cochrane collaboration in case of means and standard deviations [29] or fixed-effect meta-analysis in case of association estimates for each subgroup [32]; (3) if a study provided association estimates separately for left and right CIMT or multiple sample groups, they were combined using fixed-effect meta-analysis [32,34]; (4) if a study reported on both mean and maximum CIMT, mean CIMT was included in the analyses; and (5) if multiple publications from the same study reported on the same exposure type, the publication providing data on the longest follow-up, left and right and/or mean thickness CIMT, or with the clearest reporting or highest sample size was used.

Standardized association measures were used to pool results across studies so that the comparison of different exposure types would be possible on the same scale. The standardized mean difference (SMD) in CIMT between exposed and reference groups was used as the common association measure for binary exposure variables [31–33]. When means and standard deviations for exposed and reference were not available or could not be estimated from other statistics as reported above, SMD were estimated from the unstandardized regression coefficient and the full-sample standard deviation [33,38]. The correlation coefficient was used as the common association measure for continuous exposure variables [32,33,36,37]. When the correlation was not available, it was estimated from the regression coefficient and its standard error or the unstandardized regression coefficient and the standard deviations of the exposure and outcome variables [36,37]. If the reported association measures could not be transformed to SMD or correlation coefficient, they were analyzed and reported separately. Estimates from bivariate associations or partial associations that were least adjusted for sex were used for pooling data across studies [39]. If at least 3 studies reported on the same exposure type, with a similar definition, association estimates were pooled using the DerSimonian–Laird random-effects model. As recommended by Amrhein and colleagues [40], together with more than 800 cosignatories of the initiative "Retire statistical significance," and because we assessed multiple associations, we interpreted the point estimate as the most compatible value of the observed association. The uncertainty around the point estimate is comprised in the accompanying 95% CI, which is reported along the corresponding *p*-value.

The heterogeneity was assessed using the Cochran Q test and $I^2$ and $tau^2$ statistics [29,32,41]. $I^2$ and $tau^2$ are standard measures of heterogeneity in random-effect meta-analyses that reflect the amount of true heterogeneity between studies, i.e., the variation in effect sizes that is not due to sampling error. $I^2$ is the percentage of variation in effect sizes attributable to heterogeneity, ranging from 0% to 100%, with 0% indicating that all variation in effect sizes is due to sampling error within studies and 100% indicating that all variation is due to true heterogeneity between studies [32]. $Tau^2$ is used to assign study weights under the random-effects model [32]. Sources of heterogeneity were explored using subgroup meta-analyses, meta-regression, and sensitivity analyses. Publication bias was evaluated by the visual inspection of funnel plots and, where appropriate, by Egger test. All statistical analyses were conducted with Stata (version 16, StataCorp, Texas, USA) and Microsoft Excel (version 2019).

## Results

### Study characteristics

The searches retrieved 8,338 reports (Fig 1). Following deduplication, titles and abstracts of 6,221 reports were screened, and 154 were selected for full-text screening. A total of 50 full-text articles, pertaining to 36 studies, were eventually included. Details on study characteristics and associated reports are presented in Table 1. Studies were conducted in Europe ($n = 28$), Asia ($n = 3$), North America ($n = 3$), and Australia ($n = 2$). Some 23 studies were conducted in healthcare facilities, 6 were community/population based, 4 included a mix of the aforementioned 2 settings, and 3 did not report any information about their participant recruitment sites or sources. Exposures were evaluated in studies with a cohort ($n = 22$) or cross-sectional ($n = 14$) design. Interventions were studied in 2 randomized controlled trials. All studies

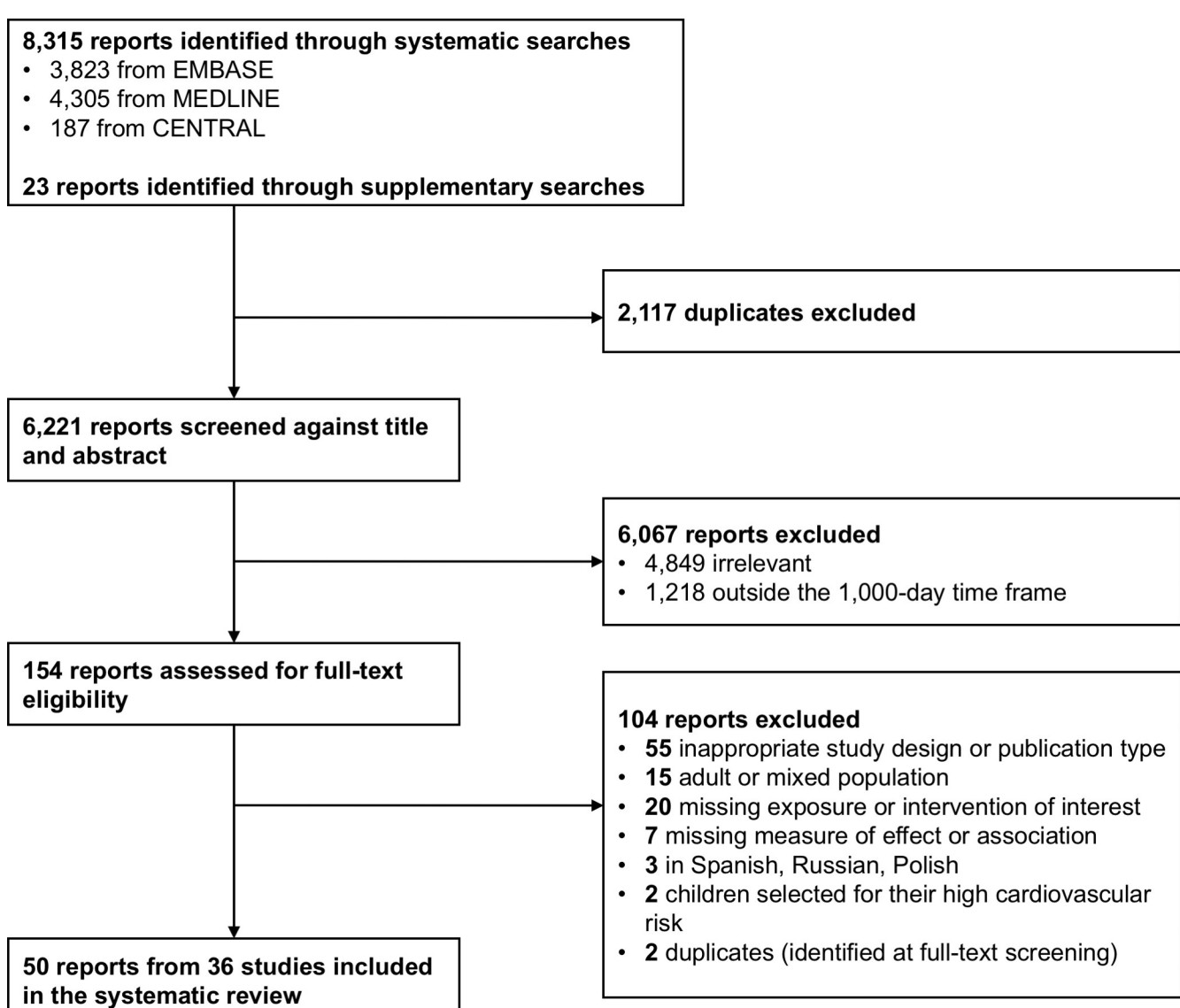

**Fig 1. Flow diagram showing the study selection process.** CENTRAL, Cochrane Central Register of Controlled Trials; EMBASE, Excerpta Medica database; MEDLINE, Medical Literature Analysis and Retrieval System Online.

**Table 1. Characteristics of included studies.**

| Author, publication year | Country | Design | Setting | N* participants | Age at CIMT assessment | Mean (SD), CIMT, mm | Exposure/intervention categories at child (1), family (2), or environmental (3) levels |
|---|---|---|---|---|---|---|---|
| **Observational studies** | | | | | | | |
| Gale and colleagues[42], 2006 | UK | cohort | healthcare facility | 216 | 9 years | right: 0.34 (0.06) | Diet and feeding practices (1), Fetal growth (1), Postnatal growth (1), Maternal weight and growth, nutrition, and physical activity (2), Socioeconomic status (3), Tobacco exposure (3) |
| Gale and colleagues [43], 2008 | UK | cohort | healthcare facility | 178 | 9 years | right: 0.34 (0.06) | Maternal weight and growth, nutrition, and physical activity (2) |
| Ayer and colleagues [44], 2009a | Australia | cohort | healthcare facility | 405 | 8 years | left and right: 0.59 (0.07) | Fetal growth (1), Pregnancy-specific factors (2), Tobacco exposure (3) |
| Ayer and colleagues [45], 2009b | Australia | cohort | healthcare facility | 405 | 8 years | left and right: 0.59 (0.06) | Diet and feeding practices (1) |
| Ayer and colleagues [46], 2011 | Australia | cohort | healthcare facility | 405 | 8 years | left and right: 0.59 (0.1) | Tobacco exposure (3) |
| Skilton and colleagues [47], 2012 | Australia | cohort | healthcare facility | 363 | 8 years | left and right: 0.59 (0.06) | Fetal growth (1), Gestational age (1) |
| Skilton and colleagues [48], 2013 | Australia | cohort | healthcare facility | 395 | 8 years | left or right: 0.77 (0.08) | Diet and feeding practices (1), Fetal growth (1), Gestational age (1), Postnatal growth (1), Socioeconomic status (3) |
| Crispi and colleagues [11], 2010 | Spain | cohort | healthcare facility | 200 | 2 to 6 years | left and right: 0.38 (0.11) | Fetal growth (1), Gestational age (1) |
| Rodriguez-Lopez and colleagues [49], 2016 | Spain | cohort | healthcare facility | 202 | 4 to 5 years | left and right: 0.38 (0.03) | Diet and feeding practices (1), Fetal growth (1), Gestational age, Other (1), Postnatal growth (1), Pregnancy-specific factors (2), Socioeconomic status (3), Tobacco exposure (3) |
| Trevisanuto and colleagues [50], 2010 | Italy | cohort | unclear | 38 | 3 to 5 years | left: 0.51 (0.08); right: 0.51 (0.07) | Fetal growth (1) |
| Evelein and colleagues [51], 2011 | the Netherlands | cohort | community/ population based | 296 | 5 years | right: 0.39 (0.04) | Diet and feeding practices (1) |
| Geerts and colleagues [52], 2012 | the Netherlands | cohort | community/ population based | 258 | 5 years | right: 0.38 (0.03) | Tobacco exposure (3) |
| Evelein and colleagues [53], 2013 | the Netherlands | cohort | community/ population based | 323 | 5 years | right: 0.39 (0.69) | Fetal growth (1), Postnatal growth (1) |
| Pluymen and colleagues [54], 2017 | the Netherlands | cohort | community/ population based | 413 | 5 years | right: 0.39 (0.04) | Diet and feeding practices (1) |
| Atabek and colleagues [55], 2011 | Turkey | cohort | healthcare facility | 55 | 48 to 72 hours | right: 0.31 (0.03) | Cardio-metabolic and inflammatory factors (1), Fetal growth (1), Pregnancy-specific factors (2) |
| Dratva and colleagues [12], 2013 | USA | cohort | community/ population based | 670 | mean age: 11 years | right: 0.57 (0.04) | Fetal growth (1), Gestational age (1), Pregnancy-specific factors (2) |
| Breton and colleagues [56], 2016b | USA | cohort | community/ population based | 240 | mean age: 11 years | N/S: 0.55 (0.05) | Epigenetics (1) |

*(Continued)*

**Table 1.** (Continued)

| Author, publication year | Country | Design | Setting | N* participants | Age at CIMT assessment | Mean (SD), CIMT, mm | Exposure/intervention categories at child (1), family (2), or environmental (3) levels |
|---|---|---|---|---|---|---|---|
| Breton and colleagues [57], 2016a | USA | cohort | community/ population based | 392 | mean age: 11 years | left: 0.56 (0.05); right: 0.57 (0.04) | Epigenetics (1), Air pollution (3) |
| Schubert and colleagues [58], 2013 | Sweden | cohort | healthcare facility | 50 | 3 months | N/S: 0.39 (0.12) | Gestational age (1) |
| Valenzuela-Alcaraz and colleagues [59], 2019 | Spain | cohort | healthcare facility | 160 | 3 years | left and right: 0.49 (0.08) | Pregnancy-specific factors (2) |
| Lee and colleagues [60], 2014 | Germany | cohort | healthcare facility | 47 | 10 to 14 years | left and right: 0.45 (0.03) | Gestational age (1) |
| Morsing and colleagues [61], 2014 | Sweden | cohort | healthcare facility | 93 | 6 to 9 years | N/S: 0.34 (0.06) | Fetal growth (1), Gestational age (1) |
| Stergiotou and colleagues [62], 2014 | Spain | cohort | healthcare facility | 201 | 0 to 1 week | left and right: 0.24 (0.05) | Cardio-metabolic and inflammatory factors (1), Fetal growth (1) |
| Gruszfeld and colleagues [63], 2015 | Belgium, Germany, Italy, Poland, and Spain | cohort | unclear | 383 | 5 years | left and right: 0.4 (0.11) | Diet and feeding practices (1), Fetal growth (1), Maternal weight and growth, nutrition, and physical activity (2), Paternal factors (2), Tobacco exposure (3) |
| Sebastiani and colleagues [64], 2016 | Spain | cohort | healthcare facility | 46 | 6 years | left and right: 0.35 (0.08) | Fetal growth (1) |
| Liu and colleagues [65], 2017 | Australia | cohort | community/ population based | 1,477 | 11 to 12 years | right: 0.58 (0.05) | Socioeconomic status (3) |
| Mohlkert and colleagues [66], 2017 | Sweden | cohort | community/ population based | 235 | 6 years | left: 0.38 (0.04); right: 0.38 (0.04) | Gestational age (1) |
| Tzschoppe and colleagues [67], 2017 | Germany | cohort | healthcare facility | 20 | 6 years | left and right: 0.38 (0.08) | Fetal growth (1) |
| Carreras-Badosa and colleagues [68], 2018 | Spain | cohort | healthcare facility | 66 | 5 to 6 years | right: 0.37 (0.03) | Maternal weight and growth, nutrition, and physical activity (2) |
| Chen and colleagues [69], 2019 | Canada | cohort | healthcare facility | 119 | 1 year | left and right: 0.56 (0.06) | Cardio-metabolic and inflammatory factors (1), Cardio-metabolic and inflammatory factors (2) |
| Prins-Van Ginkel and colleagues [70], 2019 | the Netherlands | cohort | other† | 221 | 16 years | left and right: 0.47 (0.04) | Cardio-metabolic and inflammatory factors (1) |
| Sebastiani and colleagues [71], 2019 | Spain | cohort | healthcare facility | 68 | 2 years | left and right: 0.3 (0.02) | Fetal growth (1) |
| Sundholm and colleagues [72], 2019 | Finland | cohort | other† | 201 | mean age: 6 years | left and right: 0.36 (0.04) | Maternal weight and growth, nutrition, and physical activity (2), Pregnancy-specific factors (2) |
| Jouret and colleagues [13], 2011 | France | cross-sectional | healthcare facility | 113 | mean age: 11 years | right: 0.46 (0.05) | Fetal growth (1) |

(*Continued*)

**Table 1.** (Continued)

| Author, publication year | Country | Design | Setting | N* participants | Age at CIMT assessment | Mean (SD), CIMT, mm | Exposure/intervention categories at child (1), family (2), or environmental (3) levels |
|---|---|---|---|---|---|---|---|
| Scherrer and colleagues [73], 2012 | Switzerland | cross-sectional | other[†] | 39 | mean age: 11 years | left and right: 0.39 (0.03) | Pregnancy-specific factors (2) |
| de Arriba and colleagues [74], 2013 | Spain[‡] | cross-sectional | unclear | 269 | 4 to 15 years | N/S: 0.36 (0.08) | Fetal growth (1) |
| Maurice and colleagues [75], 2014 | Canada | cross-sectional | healthcare facility | 14 | 13 to 15 years | right: 0.48 (0.05) | Fetal growth (1) |
| Xu and colleagues [76], 2014 | China | cross-sectional | healthcare facility | 124 | 3 to 7 years | left: 0.39 (0.05) | Pregnancy-specific factors (2) |
| Putra and colleagues [77], 2015 | Indonesia | cross-sectional | community/ population based | 285 | 15 to 18 years | N/S: 0.44 (0.07) | Diet and feeding practices (1) |
| Sodhi and colleagues [78], 2015 | India | cross-sectional | healthcare facility | 100 | 0 to 4 days | left and right: 0.42 (0.05) | Cardio-metabolic and inflammatory factors (1), Fetal growth (1) |
| Ciccone and colleagues [79], 2016 | Italy | cross-sectional | healthcare facility | 30 | 3 to 5 years | left: 0.51 (0.06); right: 0.52 (0.06) | Fetal growth (1), Gestational age (1) |
| Faienza and colleagues [80], 2016 | Italy | cross-sectional | healthcare facility | 52 | mean age: 10 years | left and right: 0.48 (0.06) | Fetal growth (1) |
| Olander and colleagues [81], 2016 | Finland | cross-sectional | healthcare facility | 174 | 5 to 184 hours | left and right: 0.17 (0.03) | Cardio-metabolic and inflammatory factors (1), Fetal growth (1), Gestational age (1) |
| Dilli and colleagues [82], 2017 | Turkey | cross-sectional | healthcare facility | 60 | 0 to 48 hours | left and right: 0.32 (0.03) | Cardio-metabolic and inflammatory factors (1), Fetal growth (1) |
| Stock and colleagues [14] 2018 | Austria and Italy | cross-sectional | community/ population based | 930 | 15 to 16 years | left and right: 0.38 (0.05) | Fetal growth (1), Gestational age (1) |
| Wilde and colleagues [83], 2018 | the Netherlands | cross-sectional | other[†] | 162 | 6 to 8 years | right: 0.38 (0.04) | Other (2) |
| Muñiz Fontán and colleagues [84], 2019 | Spain | cross-sectional | healthcare facility | 239 | 6 to 8 years | left and right: 0.43 (0.04) | Fetal growth (1) |
| **Interventional studies** | | | | | | | |
| Ayer and colleagues [45], 2009b | Australia | randomized controlled trial | healthcare facility | 405 | 8 years | left and right: 0.59 (0.06) | Diet and feeding practices (1) |
| Gruszfeld and colleagues [63], 2015 | Belgium, Germany, Italy, Poland, and Spain | randomized controlled trial | unclear | 383 | 5 years | left and right: 0.4 (0.11) | Diet and feeding practices (1) |

Note: Studies reporting on both interventions and exposures were analyzed as randomized controlled trial for the data on interventions and as observational cohort for the data on exposures (N = 2, namely Ayer and colleagues [45], 2009b and Gruszfeld and colleagues [63], 2015).

*N = number of participants with CIMT measurements introduced in the analyses.

[†]Mix of healthcare facility and community based.

[‡]Country of the first author.

CIMT, carotid intima-media thickness; N/S, not specified; SD, standard deviation; UK, United Kingdom; USA, United States of America.

comprised girls and boys. Overall, the minimum age at CIMT assessment was 5 hours, and the maximum age was 18 years.

## CIMT measurement methods

Image acquisition methods were relatively uniform across studies, but image analysis methods varied greatly (Tables 2 and S4). Measurements were performed on the common carotid artery (CCA) in the majority of studies (92%). However, poor or missing information on key characteristics of the CIMT measurement, namely the carotid wall, the method for edge detection and analysis of the distance between interfaces, or quality control procedures were relatively common. The segmental thickness (mean or maximum) and the type and number of sides (right, left, or the mean of both) included in the CIMT calculation were among the main sources of heterogeneity between studies. Several studies used multiple approaches for analysis; thus, up to 3 different CIMT outcomes were reported in each publication. Only 12 (33.3%) studies were at higher quality across all domains of CIMT reliability.

## Risk factors at the child level

Child risk factors evaluated across studies included (1) small size for gestational age and other fetal growth indicators, such as birth weight, length, head circumference; (2) prematurity; or (3) duration of breastfeeding. The effect of interventions targeting the dietary intake of proteins and polyunsaturated fatty acids after birth was also assessed.

CIMT was higher in children born small for gestational age (16 studies, 2,570 participants, pooled SMD 0.40 (95% CI: 0.15 to 0.64, $p$: 0.001), $I^2$: 83%) compared with those with an appropriate size at birth (Table 3; Fig 2A). The same relationship was found when analyses were restricted to children born small and with a documented prenatal diagnosis of intrauterine growth restriction (IUGR; 8 studies, 623 participants, pooled SMD 0.35 (95% CI: −0.06 to 0.77, $p$: 0.097), $I^2$: 77%). There was no clear association of prematurity with CIMT when data were pooled across all studies (7 studies, 2,024 participants, pooled SMD 0.03 (95% CI: −0.17 to 0.22, $p$: 0.805), $I^2$: 50%) (Table 3; Fig 3A). However, small size for gestational age (3 studies, 461 participants, pooled SMD: 0.64 (95% CI: 0.09 to 1.19, $p$: 0.024), $I^2$: 86%) (Fig 2B) and prematurity (3 studies, 736 participants, pooled SMD: 0.25 (95% CI: 0.02 to 0.48, $p$: 0.032), $I^2$: 0%) (Fig 3B) were consistently associated with higher CIMT, and the associations increased in magnitude when restricted to studies of higher CIMT quality. Additionally, subgroup analyses showed that CIMT was higher in infants than in older children exposed to these risk factors (Table 4). Meta-analyses of other fetal growth indicators showed that CIMT tended to be lower with higher values of birth weight or length, but the magnitude of these relationships was low (Table 3).

Inconclusive evidence was found for breastfeeding. Four studies identified a lower CIMT among children with a higher duration of breastfeeding (S5 Table). Two studies identified a higher CIMT among children exclusively breastfed compared to exclusively formula fed (S5 Table). Meta-analysis was not possible due to very heterogenous comparisons across studies. Two interventional studies on dietary interventions at the child level involved a total of 787 participants. In 1 study, a diet comprising a higher n-3 to n-6 fatty acids ratio from birth to age 5 years had no effect on CIMT (SMD 0 (95% CI: −0.20 to 0.20)) [45]. In the other study, non-breastfed children on formulas with higher protein content during their first year of life had a lower CIMT (SMD −0.21 (95% CI: −0.45 to 0.04)) compared to children on lower protein formulas [63]. Scarce evidence, reported in 1 study, existed for other factors, such as cardio-metabolic and inflammatory factors, for instance, blood pressure [81], cord blood cholesterol [55], cortisol [69], and epigenetic changes, such as DNA methylation levels following exposure to air pollution [56,57] (Tables 1 and S6).

**Table 2. CIMT measurement characteristics.**

| Author, publication year | N outcomes | Image acquisition | | | Image analysis | | | Reliability | | |
|---|---|---|---|---|---|---|---|---|---|---|
| | | Side | Segment | Wall | Edge detection, analysis of the distance between interfaces | Cardiac cycle phase | Wall thickness | Acquisition site | Analysis | Reproducibility assessment |
| Gale and colleagues [41,43], 2006, 2008 | 1 | right | CCA | far | N/S, automatic over a specific length | end diastole | N/S | higher | higher | higher |
| Ayer and colleagues [44–46], 2009a, 2009b, 2011; Skilton and colleagues[47], 2012 | 1 | left and right | CCA | far | automatic/ semiautomatic, automatic over a specific length | end diastole | mean | higher | higher | higher |
| Skilton and colleagues [85], 2013 | 1 | left or right | CCA | far | automatic/ semiautomatic, automatic over a specific length | end diastole | maximum | higher | higher | higher |
| Crispi and colleagues [11,85], 2010, 2012 | 1 | left and right | CCA | far | automatic/ semiautomatic, automatic over a specific length | end diastole | mean | higher | higher | higher |
| Rodriguez-Lopez and colleagues [49], 2016 | 1 | left and right | CCA | far | automatic/ semiautomatic, automatic over a specific length | end diastole | N/S | higher | higher | higher |
| Trevisanuto and colleagues [50], 2010 | 2 | left; right | CCA | far | manual, point-to-point measurements | N/S | maximum | higher | lower | higher |
| Evelein and colleagues [51,53], 2011, 2013 Geerts and colleagues [52], 2012 | 1 | right | CCA | far | automatic/ semiautomatic, automatic over a specific length | end diastole | N/S | higher | higher | higher |
| Pluymen and colleagues [54], 2017 | 1 | right | CCA | far | automatic/ semiautomatic, automatic over a specific length | end diastole | maximum | higher | higher | higher |
| Atabek and colleagues [55], 2011 | 1 | right | CCA | far | manual, point-to-point measurements | N/S | maximum | higher | lower | unclear |
| Dratva and colleagues [12], 2013 | 1 | right | CCA | far | automatic/ semiautomatic, automatic over a specific length | end diastole | mean | higher | higher | higher |
| Breton and colleagues [57], 2016a | 2 | left; right | CCA | far | automatic/ semiautomatic, automatic over a specific length | end diastole | N/S | higher | higher | higher |
| Breton and colleagues [56], 2016b | 1 | N/S | CCA | far | automatic/ semiautomatic, automatic over a specific length | end diastole | N/S | higher | higher | higher |
| Schubert and colleagues [58], 2013 | 1 | N/S | CCA | far | N/S, automatic over a specific length | end diastole | mean | higher | higher | higher |
| Valenzuela-Alcaraz and colleagues [59,86], 2013, 2019 | 2 | left and right | CCA | far | automatic/ semiautomatic, automatic over a specific length | end diastole | mean; maximum | higher | higher | higher |
| Lee and colleagues [60], 2014 | 1 | left and right | CCA | far | automatic/ semiautomatic, automatic over a specific length | N/S | mean | higher | higher | higher |

(*Continued*)

**Table 2.** (Continued)

| Author, publication year | N outcomes | Image acquisition | | | Image analysis | | | Reliability | | |
|---|---|---|---|---|---|---|---|---|---|---|
| | | Side | Segment | Wall | Edge detection, analysis of the distance between interfaces | Cardiac cycle phase | Wall thickness | Acquisition site | Analysis | Reproducibility assessment |
| Morsing and colleagues [61], 2014 | 1 | N/S | CCA | N/S | manual, point-to-point measurements | end diastole | mean | unclear | lower | unclear |
| Stergiotou and colleagues [62], 2014 | 2 | left and right | CCA | far | automatic/ semiautomatic, automatic over a specific length | end diastole | mean; maximum | higher | higher | higher |
| Gruszfeld and colleagues [63], 2015 | 1 | left and right | CCA | far | manual, point-to-point measurements | end diastole | mean | higher | lower | lower |
| Sebastiani and colleagues [64], 2016 | 1 | left and right | CCA | far | N/S, N/S | end diastole | N/S | higher | unclear | higher |
| Liu and colleagues [65], 2017 | 1 | right | CCA | far | automatic/ semiautomatic, automatic over a specific length | end diastole | maximum | higher | higher | higher |
| Mohlkert and colleagues [66], 2017 | 2 | left; right | CCA | far | automatic/ semiautomatic, automatic over a specific length | end diastole | mean | higher | higher | unclear |
| Tzschoppe and colleagues [67], 2017 | 1 | left and right | CCA | N/S | N/S, N/S | N/S | maximum | unclear | unclear | unclear |
| Carreras-Badosa and colleagues [68], 2018 | 1 | right | CCA | N/S | N/S, N/S | N/S | N/S | unclear | unclear | higher |
| Chen and colleagues [69], 2019 | 2 | left and right; left or right | other* | far | N/S, automatic over a specific length | N/S | N/S; maximum | unclear | unclear | unclear |
| Prins-Van Ginkel and colleagues [70], 2019 | 1 | left and right | CCA | N/S | automatic/ semiautomatic, automatic over a specific length | end distole | N/S | unclear | higher | unclear |
| Sebastiani and colleagues [71], 2019 | 1 | left and right | CCA | far | N/S, N/S | end diastole | N/S | higher | unclear | higher |
| Sundholm and colleagues [72], 2019 | 1 | left and right | CCA | far | manual, N/S | end diastole | N/S | higher | unclear | higher |
| Jouret and colleagues [13], 2011 | 1 | right | CCA | N/S | N/S, automatic over a specific length | N/S | N/S | unclear | unclear | unclear |
| Scherrer and colleagues [73,87], 2012 | 1 | left and right | CCA | N/S | automatic/ semiautomatic, automatic over a specific length | end diastole | mean | unclear | higher | higher |
| de Arriba and colleagues [74], 2013 | 1 | N/S | CCA | far | N/S, N/S | N/S | maximum | higher | unclear | unclear |
| Maurice and colleagues [75], 2014 | 1 | right | CCA | far | automatic/ semiautomatic, automatic over a specific length | end diastole | N/S | higher | higher | unclear |
| Xu and colleagues [76], 2014 | 1 | left | CCA | N/S | N/S, N/S | N/S | N/S | unclear | unclear | unclear |
| Putra and colleagues [77], 2015 | 1 | N/S | N/S | N/S | N/S, N/S | N/S | N/S | unclear | unclear | unclear |
| Sodhi and colleagues [78], 2015 | 2 | left and right | CCA | N/S | manual, point-to-point measurements | N/S | mean; maximum | unclear | lower | lower |

(*Continued*)

**Table 2.** (Continued)

| Author, publication year | N outcomes | Image acquisition | | | Image analysis | | | Reliability | | |
|---|---|---|---|---|---|---|---|---|---|---|
| | | Side | Segment | Wall | Edge detection, analysis of the distance between interfaces | Cardiac cycle phase | Wall thickness | Acquisition site | Analysis | Reproducibility assessment |
| Ciccone and colleagues [79], 2016 | 2 | left; right | CCA | far | N/S, N/S | end diastole | N/S | higher | unclear | unclear |
| Faienza and colleagues [80], 2016 | 1 | left and right | CCA | N/S | manual, point-to-point measurements | end diastole | N/S | unclear | lower | higher |
| Olander and colleagues [81], 2016 | 1 | left and right | CCA | far | manual, point-to-point measurements | end diastole | mean | higher | lower | higher |
| Dilli and colleagues [82], 2017 | 3 | left; right; left and right | CCA | far | N/S, automatic over a specific length | end diastole | N/S | higher | higher | higher |
| Stock and colleagues [14], 2018 | 2 | left and right; left or right | CCA | far | manual, point-to-point measurements | N/S | mean; maximum | higher | lower | unclear |
| Wilde and colleagues [83], 2018 | 1 | right | CCA | far | automatic/semiautomatic, automatic over a specific length | diastole | N/S | higher | higher | higher |
| Muñiz Fontán and colleagues [84], 2019 | 2 | left and right | CCA, ICA, and CB | far | automatic/semiautomatic, automatic over a specific length | end diastole | mean; maximum | higher | higher | unclear |

*Insufficient information to select a specific segment or combination of segments. Place of measurement was described as follows: "the carotid bulb, internal carotid artery, and carotid bifurcation, and at least 2 cm below the bifurcation (i.e., into the common carotid artery) were imaged. CIMT was measured over a length of 1.5 cm on each carotid artery."

CB, carotid bulb; CCA, common carotid artery; ICA, internal carotid artery; N, number; N/S, not specified.

### Risk factors at the family level

Family risk factors evaluated included (1) mode of conception; and (2) pregnancy complications, such as maternal diabetes. Children conceived through ART had, on average, an increased CIMT (3 studies, 323 participants, pooled SMD 0.78 (95% CI: −0.20 to 1.75, $p$: 0.120), $I^2$: 94%), but the imprecision around the estimate was high (Table 3; Fig 4A). There was no clear association between maternal diabetes during pregnancy and offspring's CIMT (3 studies, 658 participants, pooled SMD 0.08 (95% CI: −0.16 to 0.33, $p$: 0.495), $I^2$: 17.1%) (Table 3; Fig 4B). Scarce evidence, reported in 1 or 2 studies, existed for other risk factors, such as preeclampsia or gestational hypertension [44,49], maternal physical activity level [42], vitamin D levels [43,68], energy and macronutrient intakes during pregnancy [42], or paternal BMI [63] (Tables 1 and S5 and S6).

### Risk factors at the environmental level

Environmental risk factors evaluated included (1) exposure to tobacco (maternal, paternal, and household); and (2) socioeconomic status (maternal, family, and neighborhood). Children exposed to maternal smoking during pregnancy had, on average, an increased CIMT, but the imprecision around the estimate was high (3 studies, 909 participants, pooled SMD 0.12 (95% CI: −0.06 to 0.30, $p$: 0.205), $I^2$: 0%) (Table 3; Fig 4C). Paternal smoking during pregnancy was linked with increased CIMT in 1 study [49]. No difference in CIMT after exposure to passive smoking in the first year of life was reported in 1 study [46]. Children with a lower socioeconomic status tended to have a higher CIMT (S5 Table). Scarce evidence, reported in 1 study,

**Table 3. Summary of findings for each exposure type included in meta-analyses.**

| Category of exposure | Type of exposure | Level of comparison | N studies | N participants | Association measure | Association estimate | Heterogeneity |
|---|---|---|---|---|---|---|---|
| **Child level** | | | | | | | |
| Fetal growth | Birth size for gestational age | small* vs appropriate | 16 | 2,570 (848 vs 1,722) | SMD | 0.40 (95% CI: 0.15 to 0.64, $p$: 0.001) | $I^2$: 83%, $tau^2$: 0.18, $p$: <0.001 |
| | | small with IUGR diagnosis vs appropriate | 8 | 623 (176 vs 447) | SMD | 0.35 (95% CI: −0.06 to 0.77, $p$: 0.097) | $I^2$: 77%, $tau^2$: 0.26, $p$: <0.001 |
| | Birth weight | range of values | 7 | 1,445 | Correlation | −0.06 (95% CI: −0.19 to 0.08, $p$: 0.404) | $I^2$: 82%, $tau^2$: 0.02, $p$: <0.001 |
| | Birth length | range of values | 3 | 424 | Correlation | −0.18 (95% CI: −0.36 to 0.00, $p$: 0.052) | $I^2$: 64%, $tau^2$: 0.02, $p$: 0.062 |
| | Birth head circumference | range of values | 3 | 329 | Correlation | 0.01 (95% CI: −0.40 to 0.42, $p$: 0.960) | $I^2$: 93%, $tau^2$: 0.14, $p$: <0.001 |
| Gestational age | Gestational age | preterm† vs term | 7 | 2,024 (369 vs 1,655) | SMD | 0.03 (95% CI: −0.17 to 0.22, $p$: 0.805) | $I^2$: 50%, $tau^2$: 0.03, $p$: 0.062 |
| **Family level** | | | | | | | |
| Pregnancy-specific factors | Mode of conception | ART‡ vs natural | 3 | 323 (177 vs 146) | SMD | 0.78 (95% CI: −0.20 to 1.75, $p$: 0.120) | $I^2$: 94%, $tau^2$: 0.69, $p$: <0.001 |
| | Maternal diabetes in pregnancy | yes§ vs no | 3 | 658 (150 vs 508) | SMD | 0.08 (95% CI: −0.16 to 0.33, $p$: 0.495) | $I^2$: 17%, $tau^2$: 0.01, $p$: 0.299 |
| **Environmental level** | | | | | | | |
| Tobacco exposure | Maternal smoking in pregnancy | yes vs no | 3 | 909 (145 vs 734) | SMD | 0.12 (95% CI: −0.06 to 0.30, $p$: 0.205) | $I^2$: 0%, $tau^2$: 0, $p$: 0.380 |

*Birth weight and/or length below the 10th percentile or 2 SDs below the mean, with or without a documented prenatal diagnosis of IUGR (fetal biometry or Doppler velocimetry).

†Below 37 weeks of gestation, where specified.

‡In vitro fertilization or intracytoplasmic sperm injection.

§Gestational diabetes, where specified.

ART, assisted reproductive technologies; CI, confidence interval; IUGR, intrauterine growth restriction; N, number; p, p-value; SD, standard deviation; SMD, standardized mean difference; vs, versus.

existed for ambient air pollutants [57], such as nitrogen dioxide (NO2), ozone (O3), or particulate matter (PM10 and PM2.5) (Tables 1 and S6).

## Discussion

### Main findings

In this systematic review of 36 studies, involving 7,977 participants, multiple exposures at the child, family, or environmental levels were evaluated. Small size for gestational age had the most consistent association with increased CIMT. The magnitude of this association was higher when restricted to studies with a higher quality of CIMT measurement. The associations with conception through ART or with maternal smoking during pregnancy were not statistically significant. Evidence from interventional studies was scarce and focused exclusively on dietary interventions in the child, without showing an effect. CIMT measurement methods varied across studies, and they were frequently poorly reported, with only 33% of studies rated at higher quality across all domains of CIMT reliability.

### Comparison with other studies

To the best of our knowledge, this is the first systematic review with such a multidimensional view on risk factors in the first 1,000 days of life and CIMT in children. The underlying

## Small size for gestational age

### (A) All

| Author (Year) | N exposed | N reference | SMD (95% CI) | % Weight |
|---|---|---|---|---|
| Crispi (2010) | 80 | 120 | 0.13 (−0.15, 0.41) | 7.48 |
| Trevisanuto (2010) | 17 | 21 | 0.30 (−0.35, 0.94) | 5.24 |
| Jouret (2011) | 60 | 53 | 0.00 (−0.37, 0.37) | 6.98 |
| de Arriba (2013) | 181 | 88 | 0.66 (0.39, 0.92) | 7.59 |
| Maurice (2014) | 7 | 7 | 0.00 (−1.05, 1.05) | 3.26 |
| Morsing (2014) | 31 | 62 | −0.47 (−0.91, −0.03) | 6.56 |
| Stergiotou (2014) | 67 | 134 | 0.74 (0.44, 1.05) | 7.38 |
| Sodhi (2015) | 50 | 50 | 1.00 (0.58, 1.42) | 6.69 |
| Faienza (2016) | 27 | 25 | 1.06 (0.48, 1.65) | 5.62 |
| Olander (2016) | 39 | 90 | −0.33 (−0.71, 0.04) | 6.93 |
| Sebastiani (2016) | 27 | 19 | 0.36 (−0.23, 0.95) | 5.57 |
| Dilli (2017) | 26 | 34 | 1.15 (0.60, 1.70) | 5.82 |
| Tzschoppe (2017) | 9 | 11 | 0.95 (0.02, 1.88) | 3.72 |
| Stock (2018) | 92 | 836 | −0.08 (−0.30, 0.13) | 7.82 |
| Muñiz Fontán (2019) | 117 | 122 | 0.28 (0.03, 0.54) | 7.63 |
| Sebastiani (2019) | 18 | 50 | 1.00 (0.44, 1.57) | 5.73 |
| **Overall** | 848 | 1722 | 0.40 (0.15, 0.64) | 100.00 |

I² = 83%, p = <0.001, tau² = 0.18

### (B) Higher CIMT reliability

| Author (Year) | N exposed | N reference | SMD (95% CI) | % Weight |
|---|---|---|---|---|
| Crispi (2010) | 80 | 120 | 0.13 (−0.15, 0.41) | 36.00 |
| Stergiotou (2014) | 67 | 134 | 0.74 (0.44, 1.05) | 35.53 |
| Dilli (2017) | 26 | 34 | 1.15 (0.60, 1.70) | 28.48 |
| **Overall** | 173 | 288 | 0.64 (0.09, 1.19) | 100.00 |

I² = 86%, p = 0.001, tau² = 0.20

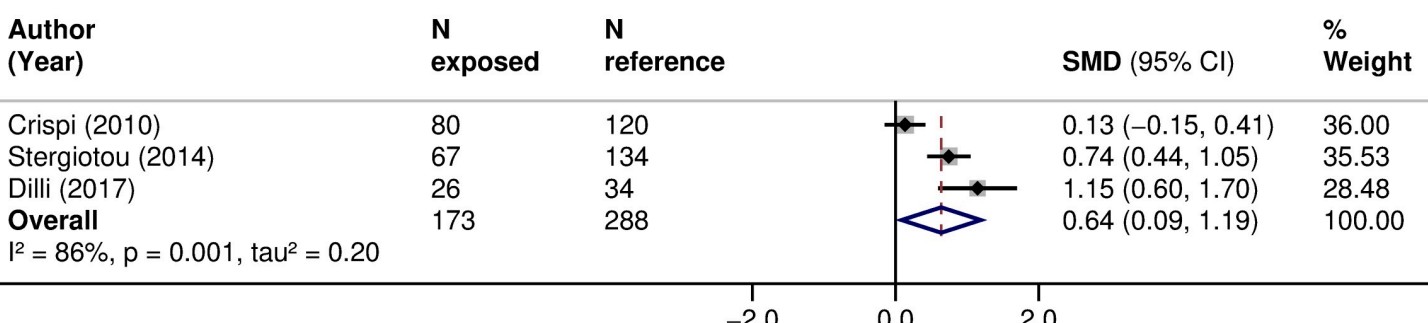

**Fig 2. Association of small size for gestational age with CIMT in children.** SMD in CIMT between children born with a small size for gestational age (exposed) and those born with an appropriate size for gestational age (reference) in (A) all studies or (B) studies at higher CIMT reliability. Weights are from random-effects model. A positive SMD corresponds to a higher CIMT in the exposed as opposed to reference. CI, confidence interval; CIMT, carotid-intima media thickness; N, sample size; p, p-value; SMD, standardized mean difference.

mechanisms of CVD programming and atherosclerosis are multifactorial. Exposure to risk factors in critical periods of development may influence life course cardiovascular health trajectories through epigenetic changes [88] and cardio-metabolic factors [89] (e.g., growth patterns,

## Prematurity

**(A) All**

| Author (Year) | N exposed | N reference | SMD (95% CI) | % Weight |
|---|---|---|---|---|
| Crispi (2010) | 40 | 80 | 0.24 (−0.14, 0.62) | 14.36 |
| Dratva (2013) | 35 | 531 | 0.35 (0.01, 0.69) | 15.99 |
| Schubert (2013) | 21 | 29 | 0.03 (−0.53, 0.59) | 8.80 |
| Morsing (2014) | 61 | 32 | −0.37 (−0.80, 0.06) | 12.46 |
| Ciccone (2016) | 15 | 15 | −0.08 (−0.80, 0.64) | 6.04 |
| Mohlkert (2017) | 114 | 121 | 0.12 (−0.13, 0.38) | 20.32 |
| Stock (2018) | 83 | 847 | −0.19 (−0.41, 0.04) | 22.03 |
| **Overall** | 369 | 1655 | 0.03 (−0.17, 0.22) | 100.00 |

I² = 50%, p = 0.062, tau² = 0.03

−2.0    0.0    2.0

**(B) Higher CIMT reliability**

| Author (Year) | N exposed | N reference | SMD (95% CI) | % Weight |
|---|---|---|---|---|
| Crispi (2010) | 40 | 80 | 0.24 (−0.14, 0.62) | 37.11 |
| Dratva (2013) | 35 | 531 | 0.35 (0.01, 0.69) | 45.83 |
| Schubert (2013) | 21 | 29 | 0.03 (−0.53, 0.59) | 17.06 |
| **Overall** | 96 | 640 | 0.25 (0.02, 0.48) | 100.00 |

I² = 0%, p = 0.627, tau² = 0

−2.0    0.0    2.0

**Fig 3. Association of prematurity with CIMT in children.** SMD in CIMT between children born preterm (exposed) and those born at term (reference) in (A) all studies or (B) studies at higher CIMT reliability. Weights are from random-effects model. A positive SMD corresponds to a higher CIMT in the exposed as opposed to reference. CI, confidence interval; CIMT, carotid-intima media thickness; N, sample size; p, p-value; SMD, standardized mean difference.

blood pressure, total cholesterol, or glucose levels), but also via direct effects on the vessels structure and function. Assessing the relationship with carotid remodeling already in children was needed to shed light on the consistency and magnitude of associations, their clinical and public health importance, as well as to provide suggestions for mechanisms that need to be addressed in subsequent high-quality studies. Given that atherosclerosis is a disease affecting the whole arterial system, similar research efforts focusing on the aortic bed are currently underway (PROSPERO; CRD42019137559).

The relationship of impaired fetal growth with a higher CVD risk in later life was among the first one to be reported in the DOHaD literature. Systematic reviews and meta-analyses in adults showed that a low birth weight is associated with an increased risk of elevated blood pressure and coronary heart disease [90]. Other studies found evidence of a thicker CIMT in young adults with a small size at birth [8,9]. We found a thicker CIMT in children that were born small for gestational age, with a stronger relationship in infants than in older children. This finding is in line with previous studies and indicates that fetal growth restriction may place the individual on a higher risk trajectory since birth. It has been argued that the definition of small for gestational age may not distinguish infants that are born small due to an intrauterine pathologic process from those that have reached their genetic potential but are constitutionally small [91]. Further, fetal growth restriction was shown to be associated with cardiac remodeling and reduced arterial compliance already in utero [92]; therefore, the severity of the growth impairment might play a key role in the vascular abnormalities of these children. Following this reasoning, we restricted our analyses to children that were born small for

**Table 4. Subgroup meta-analyses of SMD for the association of small size for gestational age or prematurity with CIMT.**

| | Birth size for gestational age (small vs appropriate) | | | | Gestational age (preterm vs term) | | | |
|---|---|---|---|---|---|---|---|---|
| | N studies | SMD | I², tau²* | p† | N studies | SMD | I², tau²* | p† |
| **All** | 16 | 0.40 (95% CI: 0.15 to 0.64, p: 0.001) | 83%, 0.18 | - | 7 | 0.03 (95% CI: −0.17 to 0.22, p: 0.805) | 50%, 0.03 | - |
| **Study design** | | | | | | | | |
| Cohort | 7 | 0.39 (95% CI: 0 to 0.78, p: 0.049) | 79%, 0.2 | 0.528 | 5 | 0.1 (95% CI: −0.13 to 0.33, p: 0.383) | 45%, 0.03 | 0.030 |
| Cross-sectional | 9 | 0.41 (95% CI: 0.07 to 0.74, p: 0.017) | 87%, 0.21 | | 2 | −0.18 (95% CI: −0.39 to 0.04, p: 0.104) | 0%, 0 | |
| **Study setting** | | | | | | | | |
| Community or population based | 1 | −0.09 (95% CI: −0.30 to 0.13, p: 0.440) | - | <0.001 | 3 | 0.08 (95% CI: −0.23 to 0.38, p: 0.626) | 73%, 0.05 | 0.716 |
| Healthcare facility | 13 | 0.43 (95% CI: 0.14 to 0.72, p: 0.004) | 83%, 0.21 | | 4 | −0.04 (95% CI: −0.34 to 0.27, p: 0.816) | 32%, 0.03 | |
| N/S | 2 | 0.60 (95% CI: 0.36 to 0.85, p: <0.001) | 1%, 0.00 | | - | - | - | |
| **CIMT wall** | | | | | | | | |
| Far | 11 | 0.38 (95% CI: 0.11 to 0.64, p: 0.005) | 82%, 0.15 | 0.756 | 6 | 0.08 (95% CI: −0.11 to 0.27, p: 0.425) | 42%, 0.02 | 0.066 |
| N/S | 5 | 0.47 (95% CI: −0.17 to 1.12, p: 0.149) | 88%, 0.46 | | 1 | −0.37 (95% CI: −0.81 to 0.06, p: 0.090) | - | |
| **CIMT sides** | | | | | | | | |
| Left and right | 12 | 0.5 (95% CI: 0.22 to 0.79, p: 0.001) | 84%, 0.19 | 0.196 | 4 | 0.02 (95% CI: −0.2 to 0.23, p: 0.889) | 41%, 0.02 | 0.059 |
| Right | 2 | 0 (95% CI: −0.35 to 0.35, p: 1.000) | 0%, 0.00 | | 1 | 0.35 (95% CI: 0.01 to 0.69, p: 0.045) | - | |
| N/S | 2 | 0.11 (95% CI: −1.00 to 1.21, p: 0.849) | 95%, 0.60 | | 2 | −0.22 (95% CI: −0.6 to 0.17, p: 0.272) | 19%, 0.02 | |
| **CIMT edge detection method** | | | | | | | | |
| Automatic or semiautomatic | 4 | 0.35 (95% CI: 0.04 to 0.67, p: 0.029) | 69%, 0.06 | <0.001 | 3 | 0.21 (95% CI: 0.03 to 0.39, p: 0.021) | 0%, 0 | 0.006 |
| Manual | 6 | 0.22 (95% CI: −0.26 to 0.71, p: 0.367) | 88%, 0.31 | | 2 | −0.23 (95% CI: −0.43 to −0.03, p: 0.025) | 0%, 0 | |
| N/S | 6 | 0.64 (95% CI: 0.27 to 1.01, p: 0.001) | 71%, 0.14 | | 2 | −0.02 (95% CI: −0.46 to 0.43, p: 0.949) | 0%, 0 | |
| **Ultrasound transducer frequency** | | | | | | | | |
| ≤12 MHz | 10 | 0.69 (95% CI: 0.45 to 0.93, p: <0.001) | 57%, 0.07 | <0.001 | 3 | 0.21 (95% CI: −0.06 to 0.48, p: 0.123) | 0%, 0 | 0.253 |
| >12 MHz | 4 | 0.03 (95% CI: −0.5 to 0.56, p: 0.904) | 90%, 0.26 | | 2 | −0.06 (95% CI: −0.66 to 0.54, p: 0.846) | 77%, 0.14 | |
| N/S | 2 | −0.06 (95% CI: −0.25 to 0.12, p: 0.505) | 0%, 0.00 | | 2 | −0.04 (95% CI: −0.34 to 0.27, p: 0.808) | 69%, 0.03 | |
| **Age** | | | | | | | | |
| 0 to 1 years | 4 | 0.63 (95% CI: −0.02 to 1.27, p: 0.057) | 91%, 0.39 | 0.002 | 1 | 0.03 (95% CI: −0.53 to 0.59, p: 0.924) | - | 0.957 |
| 2 to 16 years | 12 | 0.31 (95% CI: 0.06 to 0.55, p: 0.014) | 77%, 0.13 | | 6 | 0.02 (95% CI: −0.2 to 0.25, p: 0.828) | 58%, 0.04 | |
| **N participants** | | | | | | | | |
| <200 | 11 | 0.44 (95% CI: 0.05 to 0.84, p: 0.027) | 83%, 0.35 | 0.699 | 4 | −0.04 (95% CI: −0.34 to 0.27, p: 0.816) | 32%, 0.03 | 0.716 |
| ≥200 | 5 | 0.34 (95% CI: 0.03 to 0.65, p: 0.034) | 86%, 0.11 | | 3 | 0.08 (95% CI: −0.23 to 0.38, p: 0.626) | 73%, 0.05 | |

*I² and tau² assess within-group heterogeneity.

†p-value assesses between-group heterogeneity.

CI, confidence interval; CIMT, carotid intima-media thickness; N, number; N/S, not specified; p, p-value; SMD, standardized mean difference.

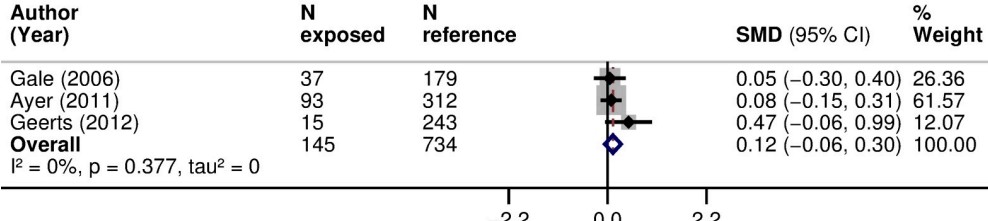

### (A) Assisted reproductive technology conception

| Author (Year) | N exposed | N reference | | SMD (95% CI) | % Weight |
|---|---|---|---|---|---|
| Scherrer (2012) | 21 | 18 | | 1.55 (0.82, 2.27) | 30.39 |
| Xu (2014) | 76 | 48 | | −0.17 (−0.53, 0.19) | 34.67 |
| Valenzuela–Alcaraz (2019) | 80 | 80 | | 1.04 (0.71, 1.37) | 34.94 |
| **Overall** | 177 | 146 | | 0.78 (−0.20, 1.75) | 100.00 |
| I² = 94%, p = <0.001, tau² = 0.69 | | | | | |

−2.3    0.0    2.3

### (B) Maternal diabetes in pregnancy

| Author (Year) | N exposed | N reference | | SMD (95% CI) | % Weight |
|---|---|---|---|---|---|
| Ayer (2009) | 24 | 381 | | 0.00 (−0.41, 0.41) | 28.84 |
| Atabek (2011) | 30 | 25 | | 0.46 (−0.07, 1.00) | 18.16 |
| Sundholm (2019) | 96 | 102 | | 0.00 (−0.28, 0.28) | 53.00 |
| **Overall** | 150 | 508 | | 0.08 (−0.16, 0.33) | 100.00 |
| I² = 17%, p = 0.299, tau² = 0.01 | | | | | |

−2.2    0.0    2.2

### (C) Maternal smoking in pregnancy

| Author (Year) | N exposed | N reference | | SMD (95% CI) | % Weight |
|---|---|---|---|---|---|
| Gale (2006) | 37 | 179 | | 0.05 (−0.30, 0.40) | 26.36 |
| Ayer (2011) | 93 | 312 | | 0.08 (−0.15, 0.31) | 61.57 |
| Geerts (2012) | 15 | 243 | | 0.47 (−0.06, 0.99) | 12.07 |
| **Overall** | 145 | 734 | | 0.12 (−0.06, 0.30) | 100.00 |
| I² = 0%, p = 0.377, tau² = 0 | | | | | |

−2.2    0.0    2.2

**Fig 4. Association of (A) ART conception, (B) maternal diabetes in pregnancy, and (C) maternal smoking in pregnancy with CIMT in children.** SMD in CIMT between children: (A) conceived through ART (exposed) or naturally (reference); (B) exposed to maternal diabetes during pregnancy (exposed) or not exposed (reference); and (C) exposed to maternal smoking during pregnancy (exposed) or not exposed (reference). Weights are from random-effects model. A positive SMD corresponds to a higher CIMT in the exposed as opposed to reference. ART, assisted reproductive technologies; CI, confidence interval; CIMT, carotid-intima media thickness; N, sample size; p, p-value; SMD, standardized mean difference.

gestational age and had a documented prenatal diagnosis of IUGR. CIMT remained thicker in IUGR children compared with those born appropriate for gestational age (Table 3). Also, the vast majority of included studies used Doppler velocimetry in the maternal or fetal vessels to diagnose IUGR; thus, these findings support the role of a well-characterized mechanism of vascular remodeling after poor fetal growth, namely the placental dysfunction [10].

The duration of gestation, maternal diabetes, or parental smoking during pregnancy are known determinants of the size at birth [1] and may also directly affect the vascular function and structure. Prematurity was shown to increase the risk of hypertension in adolescents and adults [93], probably through impaired renal development. Young adults that were born preterm were shown to have a slightly higher CIMT, noted in 3 of 4 studies of a recent systematic review and meta-analysis [94]. We did not find consistent evidence of a relationship between prematurity and higher CIMT in children. This may be because prematurity is more likely to be associated with a higher cardiovascular risk only if accompanied by fetal growth restriction or higher blood pressure in later life [8,95]. Also, it may be due to the measurement error in CIMT. In our case, it is plausible to be a combination of these mechanisms, as studies meta-

analyzed involved former preterm children, with or without poor fetal growth; therefore, a distortion of this association cannot be excluded, and CIMT was higher when restricted to studies with more reliable measurements. Of note, the relationship of CIMT with prematurity was weaker than that with small for gestational age, which pleads for a greater impact of the latter, and potentially, a more pronounced impact when the 2 risk factors are present concomitantly. Interestingly, we found inverse, but very weak, relationships of CIMT with other growth indicators, such as birth length, potentially because they do not account for the duration of gestation. We also found no clear evidence of a higher CIMT after exposure to maternal diabetes during pregnancy and a consistently higher CIMT, although of very low magnitude, in children exposed to maternal smoking during pregnancy. It is known that gestational diabetes is associated with macrosomia in the offspring, whereas maternal smoking during pregnancy is associated with poor fetal growth [1]. Our data indicate that CIMT was consistently increased in children with abnormal growth, either large or small. The effect of maternal gestational diabetes or smoking may thus be mainly indirect, through the size at birth. However, adequately powered studies that formally assess the mediator effect of size at birth on these relationships are needed to shed light on this hypothesis [96]. Finally, our finding of a higher CIMT in children conceived through ART draws attention on the highly sensitive periconceptional period, with potentially new underlying mechanisms that need to be disentangled.

Heterogeneity and poorly reported or missing information on key characteristics of the CIMT measurement methods were among the main issues that we encountered when analyzing the evidence presented herein. Similar conclusions were drawn in systematic reviews on CIMT in adults that carefully considered measurement methods when interpreting their results [6,97]. In our case, the sensitivity analyses performed for small size for gestational age and prematurity showed that the quality of CIMT measurement methods had a significant impact on the results. Additionally, CIMT values may differ with the edge detection method and the specificities of the ultrasound equipment used, such as the transducer frequency. We could notice large decreases in the magnitude of association estimates across studies using higher transducer frequencies. Our finding is in line with other evidence showing that conventional high frequencies ($\leq$12 MHz) may have insufficient ultrasound resolution and result in an overestimation of the true thickness in children under 12 years of age [18,19]. Novel techniques that measure CIMT with very high-resolution ultrasound (25 to 55 MHz) seem to be more precise in infants and young children and at very early stages of atherosclerosis [98,99].

## Strengths and limitations

The strength of our systematic review lies primarily in the fact that it was performed according to a detailed protocol that was prospectively registered on PROSPERO and published in full in a peer-reviewed journal. Secondly, we collected extensive data on the CIMT measurement characteristics, which permitted interpretation of results in the light of the quality of measurement. Thirdly, we included both observational and interventional studies reporting on exposures or interventions at the child, family, or environmental levels. This helped us have a comprehensive overview on the current status of the evidence and highlight needed areas of future research.

The main limitation is that the body of evidence for this systematic review is largely observational, which is of low certainty by design and restricts the ability to draw conclusions about causality. Additionally, the degree of confidence in results is limited by the high between-study heterogeneity. Studies varied on several aspects, such as design, exposure metric, age at CIMT assessment, or CIMT image analysis. The variability in age at CIMT assessment was large, from 0 to 18 years. Our subgroup analysis was mainly based on the age range reported in each

study and, thus, could only distinguish the infants from the other pediatric age groups combined. Meta-analyses were performed mainly using estimates from bivariate associations. Data transformation and simplification were needed, and associations were estimated from various statistics so that relationships could be summarized in a consistent way across studies. Therefore, differences in body size between exposed and nonexposed could not be taken into account in the analyses, confounding bias cannot be excluded, and our association estimates may be distorted, probably underestimated, due to measurement error in the CIMT. Furthermore, CIMT was evaluated in youth with tiny, submillimetric vessel structures, and many studies had relatively small samples and few individuals exposed to the risk factors of interest, which resulted in large CIs for the estimates. The low number of studies meta-analyzed often impeded conduction of subgroup analyses to explore the sources of heterogeneity predefined in the protocol. However, to increase our confidence in the overall estimates, whenever possible, we conducted analyses restricted to studies at higher quality for all CIMT reliability domains. Other limitations are represented by exclusion of studies not reported in English or French [100–102], publication bias, and selective exposure or analysis reporting. Usable data for the majority of exposures or interventions were reported by a low to a very low number of studies. However, for small size for gestational age, the risk factor most studied, the funnel plot inspection, and Egger test were not indicative of publication bias (S2 Fig).

## Implications for research and practice

We highlight the need of a standardized ultrasound protocol for measuring CIMT in children. Due to the paucity of interventional studies and the limited quality of CIMT measurements, further high-quality studies are needed to justify the use of CIMT for child CVD risk assessment in clinical practice. However, promotion of a healthy lifestyle is important at any age, and screening of postnatal CVD factors, such as hypertension or obesity, targeted at children exposed to risk factors in the first 1,000 days of life, may be warranted.

Further, our study indicated that children that were small for gestational age had a higher CIMT in early life. Prevention of small for gestational age could have a big impact from a public health perspective. Its prevalence varies among populations and rises with decreasing gestational age and availability of resources, reaching about 20% in low and middle-income countries [103]. Moreover, modifiable factors, including gestational smoking or poor maternal nutrition, account for a large share of small for gestational age infants [104]. In fact, smoking cessation interventions during pregnancy were shown to result in a 17% reduction in low birth weight cases [105]. Public health strategies to reduce smoking and improve the nutritional status in women of reproductive age should be advocated and may improve the cardiovascular health too [105,106].

## Conclusions

In our meta-analyses, we found several risk factors in the first 1,000 days of life that may be associated with increased CIMT in infants, children, and adolescents. Poor fetal growth had the most consistent and strongest association with increased CIMT. The associations with ART conception or with maternal smoking during pregnancy were not statistically significant. From a DOHaD perspective, exposure to adverse experiences in early life determines adaptative responses in the organs' structure and function with lifelong effects on cardiovascular health and disease, ranging from elevated blood pressure, early vascular aging and increased CIMT, and premature CVD morbidity and mortality [1,107]. Therefore, assessing CIMT early in life is important to distinguish between changes occurring in childhood and adulthood, with the goal of better characterizing lifetime risk trajectories. Provided this hypothesis will stand the test of time and effective intervention, the interplay of risk factors in the first 1,000

days of life and vascular remodeling will offer a great opportunity for primordial prevention of CVD [108].

## Supporting information

**S1 PRISMA Checklist. Preferred Reporting Items for Systematic Reviews and Meta-Analyses.**
(PDF)

**S1 Fig. Assessment of study quality for each exposure type included in meta-analyses.**
(PDF)

**S2 Fig. Assessment of small-study effects, including publication bias, for each exposure type included in meta-analyses.**
(PDF)

**S3 Fig. Random-effects meta-regression examining the influence of sample size on the association of small size for gestational age with CIMT.**
(PDF)

**S1 Table. Strategies for systematic searches.**
(PDF)

**S2 Table. Strategies for supplementary searches.**
(PDF)

**S3 Table. Criteria for CIMT quality assessment.**
(PDF)

**S4 Table. CIMT equipment and operator.**
(PDF)

**S5 Table. Association of CIMT with main exposures or interventions types in the first 1,000 days of life.**
(PDF)

**S6 Table. Other exposure types in the first 1,000 days of life by level and category of exposure (reported in a single study).**
(PDF)

**S7 Table. Criteria for study quality assessment.**
(PDF)

**S8 Table. Assessment of study quality for each intervention type in interventional studies.**
(PDF)

**S9 Table. Assessment of study quality for observational studies included in meta-analyses.**
(PDF)

## Acknowledgments

The authors thank Thomas Brauchli (Data and Documentation Unit, Center for Primary Care and Public Health (UNISANTÉ), University of Lausanne, Lausanne, Switzerland) and Cécile Jaques (Medical Library, Lausanne University Hospital, University of Lausanne, Lausanne, Switzerland), librarians, who assisted with the development of the literature search strategies and provided technical advice for running the search queries in the databases.

## Author Contributions

**Conceptualization:** Adina Mihaela Epure, Arnaud Chiolero, Nicole Sekarski.

**Data curation:** Adina Mihaela Epure.

**Formal analysis:** Adina Mihaela Epure, Magali Rios-Leyvraz, Daniela Anker, Bruno R. da Costa, Arnaud Chiolero.

**Funding acquisition:** Arnaud Chiolero.

**Investigation:** Adina Mihaela Epure, Magali Rios-Leyvraz, Daniela Anker, Stefano Di Bernardo, Arnaud Chiolero, Nicole Sekarski.

**Methodology:** Adina Mihaela Epure, Stefano Di Bernardo, Bruno R. da Costa, Arnaud Chiolero, Nicole Sekarski.

**Project administration:** Adina Mihaela Epure, Arnaud Chiolero.

**Supervision:** Bruno R. da Costa, Arnaud Chiolero, Nicole Sekarski.

**Visualization:** Adina Mihaela Epure.

**Writing – original draft:** Adina Mihaela Epure.

**Writing – review & editing:** Adina Mihaela Epure, Magali Rios-Leyvraz, Daniela Anker, Stefano Di Bernardo, Bruno R. da Costa, Arnaud Chiolero, Nicole Sekarski.

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
