## [Editor Report · Decision Letter 0]

11 May 2020

Dear Dr Epure, 

Thank you for submitting your manuscript entitled "First 1000 days risk factors for carotid intima-media thickness in childhood: a systematic review with meta-analyses" for consideration by PLOS Medicine.

Your manuscript has now been evaluated by the PLOS Medicine editorial staff and I am writing to let you know that we would like to send your submission out for external peer review.

Kind regards,

Artur Arikainen,

Associate Editor

PLOS Medicine

---

## [Decision Letter · Decision Letter 1]

20 Jun 2020

Dear Dr. Epure,

Thank you very much for submitting your manuscript "First 1000 days risk factors for carotid intima-media thickness in childhood: a systematic review with meta-analyses" (PMEDICINE-D-20-01898R1) for consideration at PLOS Medicine. 

[LINK]

In light of these reviews, I am afraid that we will not be able to accept the manuscript for publication in the journal in its current form, but we would like to consider a revised version that addresses the reviewers' and editors' comments. Obviously we cannot make any decision about publication until we have seen the revised manuscript and your response, and we plan to seek re-review by one or more of the reviewers. 

We expect to receive your revised manuscript by Jul 13 2020 11:59PM. Please email us (plosmedicine@plos.org) if you have any questions or concerns.

We look forward to receiving your revised manuscript. 

Sincerely,

Emma Veitch, PhD

PLOS Medicine

On behalf of Clare Stone, PhD, Acting Chief Editor,

PLOS Medicine

plosmedicine.org

*In the last sentence of the Abstract Methods and Findings section, please add a brief note commenting on any key limitation(s) of the study's methodology.

Comments from the reviewers:

Reviewer #1: See attachment

Michael Dewey

Reviewer #2: This is a systematic review and meta-analysis titled "First 1000 days risk factors for carotid intima media thickness in childhood: a systematic review and meta-analysis" assessing different DOHaD variables associations with CIMT mainly during childhood (<10y). The protocol has been published previously (Epure AM, Leyvraz M,Mivelaz Y, et al. Risk factors and determinants of carotid intimamedia thickness in children: protocol for a systematic review and meta-analysis. BMJ Open 2018;0:e019644). The focus from the original protocol (pediatric age group 0-18; postnatal child CVD risk factors obesity, hypertension, smoking, disease and treatments) has changed in the current review manuscript to focus on younger pediatric age and the role of early life DOHaD predictors. 

The title uses the term "risk factors for CIMT in childhood", but as there is currently no evidence linking CIMT in infancy/childhood to CVD disease later in life, I would certainly propose a more comprehensive vascular phenotype (not only CIMT) in children. Furthermore, if the main focus and aim is to explore relations between DOHaD and vascular changes in children, one would perhaps expect a more comprehensive set of vascular variables included (other sites of intima-media thickness, different measures of arterial function such as regional and local stiffness, blood pressure and heart rate, perhaps also endothelial function for older children) and a more comprehensive set of covariates on postnatal anthropometrics, growth, body composition, and different traditional aspects of CVR (diet, activity etc.). The present manuscript is, thus, limited to child CIMT only, and mainly prenatal, perinatal factors and early life DOHaD factors. In addition, the high heterogeneity and apparent problems in the reported CIMT measurement methodology in infancy/childhood makes the analysis with DOHaD variables and conclusions thereof challenging and uncertain indeed.

The CIMT methodology issues warranted the investigators to critically evaluate and explore explanations for this variance, as mentioned in the published protocol as well. This is warmly endorsed by this reviewer, and forms a major strength of this manuscript. The manuscript provides in-depth analyses to shed light on factors related with this heterogeneity and how this might influence the DOHaD analyses. The current guidelines on vascular assessments and reports of population and healthy CIMT references in the pediatric age group have been made to address mainly adolescent populations with CVR and different disease states and treatments affecting the vasculature. This review is welcome in attempting to provide a summary of the very diverse current literature of the ultrasound CIMT during infancy and childhood.

The manuscript reads well, many tables and figures are included and the authors have made good attempts in trying to address the objectives. The review also seems to include the most relevant articles in the field. There are, however, some concerns related with method presentation and the analyses and the conclusions that warrant commenting and modifications to the presentation, as outlined in the comments to the authors.

1. Title and abstract. See general commenting and suggestion above for revising wording in title. Mean difference 0.40. Please provide information on adjustments for body size, HC size, gestational age, child age made in this main result, as appropriate. The first sentence in the conclusion: "Several risk factors in the first 1000 days of life may be associated with increased CIMT during childhood", comments on ART and smoking (tendencies only) also seem poorly substantiated. What intervention?

2. Introduction. Please see general commenting above. The intro could focus more on the objective, i.e. 1) relations with DOHad and 2) challenges with ultrasound derived pediatric IMT in order to provide a rationale for the study. The second paragraph could be more focused on the main objective. What is known about the role of early life factors, early child vascular phenotype assessment, and later CVD, and why should the vascular phenotype be assessed at this stage? The last sentence relating to the guidelines could emphasize the difference in age group addressed in the guidelines compared with the focus of the present manuscript.

3. Methods. Methods/Eligibility criteria/Participants. Definition of child population. "Apparently healthy children" vs children with different disease states? On the other hand "Both apparently healthy children and individuals with clinical conditions will be included." (information from PROSPERO-registry). And furthermore in published study protocol: "We will include studies in children from birth up to 18 years. Studies with both children and adults will be included if the data for children can be extracted separately. Both apparently healthy children and subjects with clinical conditions will be included" The manuscript, however, states, "Studies in children with postnatal conditions that confer an increased risk for CVD, e.g. hypertension or diabetes, were excluded. Additionally, children with rare conditions (e.g. congenital heart disease) were not included as they are not representative of the general population". Both these exclusion criteria seem irrational as future cardiovascular disease is of major interest in and early development and interventions performed could well alter both. In addition, the study includes analyses related with maternal gestational risk (egg gestational diabetes), premature birth, abnormal fetal growth, that are per se risk factors for later disease development in offspring. What is the rationale for these exclusion statements? Figure 1 shows that adult populations were excluded. Study population could then be better defined in methods and mentioned in title/abstract as well to avoid confusion.

4. Childhood is per se not defined as 0-18y, but as the age span ranging from birth to adolescence. On the other hand although the study states that they included children aged 0-18. Table 3 shows that a majority of the study subjects included belong to the younger age groups (36/50 studies report age less than 9y), so the focus is on childhood. However, the use of the term childhood should be consistent with common definitions. Please check this the use of this terminology throughout manuscript including title, abstract, manuscript text, figure and tables, supplements.

5. Comparators. No eligibility criteria is not provided although this seems key in the comparison. Published study protocol states: "Studies assessing specific clinical conditions in relationship with CIMT will be included provided they use a control group without the clinical condition of interest". Age, body size, puberty, CVR-profile including BP, race etc. are well known important key predictors of CIMT in children. Do the authors mean to say that no requirements for comparator groups needs to be specified in order to assess differences in CIMT in relation to DOHaD variables? Differences in vascular dimensions reported in the different studies should be adjusted for differences in key covariates. This is not clearly presented in Data Analysis. Please do elaborate and revise accordingly.

6. Exposures/interventions. "For observational studies, we considered for inclusion prenatal and postnatal exposures, at the individual, familial, or environmental level, which reflected the developmental milieu of the child and occurred between conception and age 24 months" This relates to the DOHaD Figure 1 states that 20 articles were excluded due to missing exposure or intervention of interest. Please specify in more clear terms what prenatal and postnatal exposures were considered here? Birth weight, gestational age at birth, maternal disease, gestational disease or complication, infections, medications etc. Examples could be provided.

7. Disagreement between reviewers were solved by discussion. Could this be presented with numbers somehow?

8. Data extraction. Quality of study assessment. Supplemental Table 3. Criteria for study quality and study reliability tool criteria and algorithm of judgement seems arbitrary and poorly justified. Why is at least 2 images/frames needed in addition to manual tracings to indicate improved reliability? How does assessing more frames improve CIMT measurement reliability? No data regarding this is presented. Intra and interobserver variabilities are commonly performed in standardized vascular laboratories as quality assurance measures, and may not always be reported in articles methods section. The non-reporting of these (or only referring to a reference) is then by no means evidence of a lower technical operator performance or lower measurement reproducibility. There other more important factors related with this that could be included: standardized imaging and measurement protocols, operator experience and skill, image quality, ultrasound equipment and transducer frequency, imaging feasibility (child preparation with measurements performed at rest; sedation as appropriate), site of carotid intima media thickness, just to mention a few. Reading Box 2 in the published study protocol in BMJ Open suggests a different approach for this assessment. Please justify the choice of different factors included in this manuscript Supplemental table 3.

Moreover, "The quality of each study was evaluated using an adjusted version of the Newcastle-Ottawa Scale for observational studies and Cochrane's collaboration risk of bias tool for experimental studies." How this was done remain largely unclear. This also applies to the information provided in Supplemental Table S7, Supplementary Figure 1S. How was study quality data generated? What defines a low quality study and a high quality study addressed in subanalyses and mentioned in the discussion? The study quality assessment should be clearly presented and outlined.

9. Data analyses. The SMD measure adjusts for methods differences in scale and precision. However, there may be differences between the exposure and comparator that are explained by covariates such as body size (weight, lean body mass or head circumference) that is not accounted for in this meta-analysis metric. The main stated results relates to being born small for gestational age. How differences in body size between exposure and comparator was (reference) groups taken into account in the generation of the SMD and this conclusion? How does measurement precision affect SMD (influence of the SD on the SMD metric?) and how does this influence the reported outcome SMD and conclusions thereof? 

10. The reported I2 50-83% (child level) in conjunction with the SMD in Table 3 indicate a high heterogeneity and inconsistency between the studies included in the meta-analysis (similarly high in Table 4 subgroup analyses). Could the high heterogeneity be explained by methodological differences between the studies? Could this be explored in more detail. Moreover, how does the high heterogeneity between the studies impact on the conclusion of a higher CIMT among children born small for gestational age? Please explain the use of I2 and tau2 metrics of heterogeneity in Data Analysis and, in particular, include interpretations on these as well. In addition, please elaborate on this in the manuscript discussion.

11. Results. Table 4, footnote. "P-value reported for between group heterogeneity". What groups? Is this p-value in conjunction with I2 and tau2 metrics (similar to table 3) between small vs appropriate birth size (weight?) or is it some kind of comparison of the heterogeneity related with subgroup factor such as transducer frequency or age? Please clarify.

12. Table 4 shows that the higher CIMT in children born small for gestational age (SMD) disappears when assessing studies using higher ultrasound frequencies. This finding should be highlighted, as the measurement accuracy is likely to be much better younger populations with thinner CIMTs.

13. Exposure to maternal smoking had a consistent association with higher CIMT (SMD 0.12, CI95% -0.06-0.30), is this statistically significant? Is this mediated by the effect of gestational smoking on birth weight?

14. Figure 2. Small size for gestational age. What was the criteria for SGA applied? What was the age of the population at CIMT assessment? Are these newborns or children of higher age? What is the effect of postnatal growth? Are the exposure and reference groups comparable with respect to age and body size?

15. What is number in brackets after first author name in Supplemental Table S4?

16. Supplemental Table 5. Birth size (weight). How comparable are the studies concerning birth size? Has the mean difference been adjusted for body size? Covariates? Is the CIMT in accordance with body size (lean body mass or HC size)?

17. Discussion. Discussion. … followed by ART and maternal smoking during pregnancy consistent effect on CIMT? Was the effect of ART statistically significant? Reported as a tendency in results (SMD 0.78, CI95% -0.20 to 1.75).

18. Please address CIMT associations with race and sex, and body composition (e.g. adiposity)? 

19. What does the SMD 0.40 correspond to in mm? What is the clinical relevance of this from a CVR perspective?

20. Overall, the discussion is too long and speculative. Please focus on discussing in relation to objective. The recommendations for CIMT measurements presented in the conclusions are not warranted by the results, they are too strong and very subjective (e.g. no data provided suggesting superiority of border detection software in pediatric CIMT or the analysis of 2 image frames using calipers), the line of reasoning is missing here.

Minor comment from reviewer related with CIMT methodology

The authors might be interested in studies addressing non-invasive ultrasound-derived pediatric (and adult) arterial layer method validation using histology (Sarkola et al Atherosclerosis. 2010 Oct;212(2):516-23; Sarkola et al Atherosclerosis. 2012 Sep;224(1):102-7; Sundholm Atherosclerosis. 2015 Apr;239(2):523-7; Sundholm et al Ultrasound Med Biol. 2019 Aug;45(8):2010-2018). These studies illustrate important caveats related with the ultrasound CIMT measurement during infancy/childhood. In brief, these are related with 1) the true histological thinness of the arterial wall layers during infancy/childhood exemplified in the present review by references reporting ultrasound IMTs in infants (e.g.Trevisanuto et al 2010, Schubert et al 2013, Sodhi et al 2015, Chen et al 2019) similar to commonly reported in young adult age (e.g. Engelen et al https://doi.org/10.1093/eurheartj/ehs380), and references reporting IMTs childhood (e.g. multiple studies by Ayer and Skilton, Ciccone et al 2016, Valenzuela-Alcaraz et al 2019) similar to commonly reported in older adult age which is biologically impossible, 2) frequency dependence of ultrasound axial resolution (i.e. lower frequency providing higher IMT) and ultrasound frequency effect on IMT measurement accuracy (conventional frequency axial resolution limit >0.25 mm higher than infancy/childhood IMT <0.25). I hope that these could guide the authors in their revision.

Review provided by Dr Taisto Sarkola.

Reviewer #3: This meta-analysis summarizes the relevance of perinatal life factors for carotid-intima thickness in children and adolescents. In view of the fact that the first 1000 days are regarded as a potentially critical period for shaping later cardiometabolic risk this is a topic of substantial public health relevance. In addition, evidence from several studies suggests that associations of factors acting in this window of opportunity may already impact carotid intima-media thickness, i.e. an established marker of subclinical atherosclerosis.

Nonetheless, a number of issues arise with this manuscript

Major

* Main results. Only one of the three "main" results (born small for gestational age) is statistically significant. This is not correctly presented and discussed by the authors. 

* Aim 2: 

o The necessity for cIMT method appraisal remains elusive (I do not question them, but they should be explained in the introduction). Why should the between-study heterogeneity matter? What are the main problems associated with this and could this affect the associations under investigation?

o The authors provide very little information on their cIMT measurement quality assessment. 

o Generally, the term "reliability" appears incorrect. Many of the aspects assessed could also affect the validity of the measurements.

* Intervention studies:

o These can only be considered as such if the primary endpoint of the study was cIMT. This was only the case for the study of Ayer et al. (Ref. 54) The second study had body composition as a major outcome. Hence, the post-hoc analysis of this study for an association with later cIMT should be regarded as an observational cohort study.

o The study of Ayer et al. (Ref. 54) is mentioned both as a cohort and an intervention study. Please, clarify.

* Exposures

o it remains elusive how the authors arrived at their set of exposures. Were there any a priori criteria (they were not part of the search strategy)? Were all exposures analysed in a relevant publication included?

o Many exposures could have multiple indicators, e.g. duration of breastfeeding would certainly be reported more specifically (exclusive vs full vs any), yet it is not clear how the authors attempted to address/combine these.

o A Table on all early life risk factors and their "definitions" would be helpful, e.g. clarify "diet and feeding practices"

o Some of the exposures appear questionable: epigenetic changes, cortisol and cardiometabolic factors in childhood are presumably markers of mechanisms by which perinatal risk factors could affect the cardiometabolic outcome cIMT rather than exposures

* Age at outcome

o One of the main limitations is the large variability in the age at outcome (0-18 years). This is hardly mentioned and the sensitivity analysis only distinguishes 0-1 from 2-18 years.

* Search

o End date March 2019 - this is outdated. Please, update.

o Were searches performed in duplicate?

* Risk of bias / Grading of evidence

o The authors use the Newcastle Ottawa scale for observational studies, but do not report findings per study. They only provide a summary figure by exposure, i.e. the reader cannot confirm their judgement

o Some aspects in this figure would need explanation (i.e. "same method of ascertainment or "comparability") 

o The Assessment of study quality for interventions studies given in Supplementary Table 7 is not referred to in the text and the table is not intelligible. Does "high" indicate high quality or high risk of bias?

o The quality of meta-evidence needs to be graded

* Discussion of the findings

o An appraisal of the "effect sizes" and their public health relevance is missing

o Heterogeneity due to the age-range at outcome measurement (only first year vs 2-18 yrs) longitudinal vs cross-sectional needs to be discussed

o Lack in focus: The authors discuss a number of potential mechanisms driving their observations, even for small sensitivity analyses for which data are not shown (and are perhaps not significant). In view of this reviewer, this should be focussing on the main findings of the meta-analysis only.

Minor

* Abstract: CIMT methods cannot be appraised with a meta-analysis

* Number for observational studies should be given separately for cross-sectional and longitudinal studies 

* Why was this study done: "It is unclear whether …" is not a convincing argument to perform such a study. Preliminary (mechanistic) evidence suggesting that this could be relevant should be given.

* Funding information should not be mentioned in methods and findings

* What does the age range in the abstract refer to? Age at cIMT measurement?

* The authors should already introduce the concept of perinatal programming and the mechanisms discussed in this context in the background. Why should we expect that early life factors could already affect cIMT in childhood/adolescence?

* The conclusion on CIMT method "cIMT measurement protocols are heterogeneous and often poorly reported" is vague. What does this refer to? Validity or reliability of the method? Lack of data on intra- and inter-observer variability? Potential for investigator-associated bias (unstandardized measurement procedure)? 

* Purpose of a meta-analysis. In the discussion the authors state that their review is needed to better understand DOHaD mechanisms. In the view of this reviewer a meta-analysis based primarily on observational studies can only provide suggestions for potential mechanisms which need to be addressed in other subsequent studies. Instead, quantification of associations, evidence on public health relevance etc. are most likely more important contributions of a meta-analysis. 

* Experimental studies should be termed intervention studies

* Participants cannot be recruited pre-birth, whereas their mothers can

* Please check language, e.g. lines 148-150, does not appear to be a sentence

* Results: When referring to specific studies the reference should be given

* Line 293: this relation is no longer significant!

* Strength: performing a meta-analysis according to a protocol is not a strength. 

* How was the association between study quality (RoB) and method quality?

* Strength of relationships (e.g. line 301) are an interpretation of the results, not a result per se

* Fixed-effects meta-analysis was used for data simplification, yet random-effects meta-analysis for the main analysis. Please explain.

* The discussion of the breastfeeding non-finding is misleading and should be shortened. The findings in this meta-analysis do not permit any public health statements relating to breastfeeding.

[LINK]

---

## [Decision Letter · Decision Letter 2]

17 Sep 2020

Dear Dr. Epure,

Thank you very much for re-submitting your manuscript "First 1,000 days risk factors for carotid intima-media thickness in infants, children, and adolescents: a systematic review with meta-analyses" (PMEDICINE-D-20-01898R2) for review by PLOS Medicine.

I have discussed the paper with my colleagues and the academic editor and it was also seen again by three reviewers. I am pleased to say that provided the remaining editorial and production issues are dealt with we are planning to accept the paper for publication in the journal.

[LINK]

We look forward to receiving the revised manuscript by Sep 24 2020 11:59PM. 

Sincerely,

Artur Arikainen, 

Associate Editor 

PLOS Medicine

plosmedicine.org

Requests from Editors:

1. Please address the reviewers’ final comments below.

2. Title: Please amend to: “Risk factors for carotid intima-media thickness during first 1,000 days of life in infants, children, and adolescents: a systematic review with meta-analyses”

3. Abstract: 

a. Line 33: Please include the search date ranges.

b. Please include the measure of study quality/bias and the overall quality or risk of bias of the included studies.

c. Please include a list of the risk factors/exposures tested.

d. Line 44: For ART and maternal smoking, we ask that you rephrase this to something like: "...non-significantly increased CIMT…" for these associations. Similarly, it may be necessary to rephrase lines 52-54 and 71-74, which currently refer to associations with several risk factors:

i. “In our meta-analyses, we found that several risk factors in the first 1,000 days of life may be associated with increased CIMT during childhood. Poor fetal growth had the most consistent relationship with increased CIMT, while for ART conception and smoking during pregnancy the increase was not significant.”

ii. “Risk factors in the first 1,000 days of life relating to poor fetal growth are associated with increased CIMT in infants, children, and adolescents. Conception through assisted reproductive technologies and maternal smoking during pregnancy also increased CIMT, but non-significantly.”

e. Please quantify all results with p values as well as 95% CI. Please include data for the following: “When restricted to studies of higher quality of CIMT measurement, the relationships were stronger.”

f. Please mention the primary funding source.

4. Author Summary:

a. Lines 65-66: Please clarify for a lay audience the significance of “vascular morphology” and “increased carotid intima-media thickness”. Perhaps also define “vascular”.

b. Line 69: Please clarify “…systematic review of published literature…”

c. Line 70: CIMT is not defined earlier.

d. Line 83: Please clarify or define “primordial” in this context.

5. Line 120: We recommend removing this subheading.

6. When completing the PRISMA checklist, please use section and paragraph numbers, rather than page numbers.

7. Lines 131-132: Please cite the included PRISMA Checklist as “S1 Checklist” here.

8. Please remove spaces from within citation callouts, eg. (line 535) “…health too [103,104].”

------------------

Comments from Reviewers:

Reviewer #1: The authors have addressed my points.

I still find it regrettable that studies published in languages other than English or French were deliberately excluded and I suggest that the references to the three that were found should be included somewhere for the benefit of future researchers who can understand them.

Michael Dewey

Reviewer #2: To the editor and authors,

For this reviewer the comments have been very well addressed in the responses and the manuscript overall. The authors have made an extraordinary effort in the systematic analyses and meta-analyses of this review. In this reviewers view, the responses to the comments provide essential information to the manuscript and, therefore, the publication of this document is warmly recommended in conjunction with the manuscript.

This reviewer has only on additional comment to provide for the revised version. This relates to the main conclusion of CIMT being increased due to poor fetal growth (or born small for gestational age) with analyses provided in Table 4, Table 5S, Results lines 315-327 and Fig 2. 

Differences in CIMT attributed to differences in body size/anthropometrics has not adequately been taken into account in the analyses in response the reviewer comments 9 and 16. The following responses are provided by the authors in relation to this issues: Response 9 (differences attributed to body size/anthropometrics ignored in the response), and Response 16 "…CIMT was measured in mm, thus, it was not normalized according to body size. "

It is well known that the most important determinant of cardiovascular dimensions in the growing fetus and child is body size and anthropometry. Differences related with e.g. brain sparing in the setting of IUGR and differences in e.g. head circumference (HC) has not been adequately addressed in this review (although partly for HC in Table 5S). This review does not then answer the question: Is CIMT proportional to head/brain anthropometrics in poor fetal growth/SGA?

The lack of analyses accounting for differences in HC/brain in assessing CIMT in relation to poor fetal growth/SGA is then a limitation (may not be adequately reported in the original references too precluding these analyses) in the analyses and this should briefly be acknowledged in the Discussion/limitations section.

Taisto Sarkola

Reviewer #3: The authors have responded to my comments in detail

[LINK]

---

## [Editor Report · Decision Letter 3]

19 Oct 2020

Dear Dr Epure, 

On behalf of my colleagues and the academic editor, Dr. Sanjay Basu, I am delighted to inform you that your manuscript entitled "Risk factors during first 1,000 days of life for carotid intima-media thickness in infants, children, and adolescents: a systematic review with meta-analyses" (PMEDICINE-D-20-01898R3) has been accepted for publication in PLOS Medicine. 

PRODUCTION PROCESS

Before publication you will see the copyedited word document (within 5 business days) and a PDF proof shortly after that. The copyeditor will be in touch shortly before sending you the copyedited Word document. We will make some revisions at copyediting stage to conform to our general style, and for clarification. When you receive this version you should check and revise it very carefully, including figures, tables, references, and supporting information, because corrections at the next stage (proofs) will be strictly limited to (1) errors in author names or affiliations, (2) errors of scientific fact that would cause misunderstandings to readers, and (3) printer's (introduced) errors. Please return the copyedited file within 2 business days in order to ensure timely delivery of the PDF proof. 

If you are likely to be away when either this document or the proof is sent, please ensure we have contact information of a second person, as we will need you to respond quickly at each point. Given the disruptions resulting from the ongoing COVID-19 pandemic, there may be delays in the production process. We apologise in advance for any inconvenience caused and will do our best to minimize impact as far as possible.

PRESS

PROFILE INFORMATION

Thank you again for submitting the manuscript to PLOS Medicine. We look forward to publishing it. 

Best wishes, 

Artur Arikainen, 

Associate Editor 

PLOS Medicine

plosmedicine.org